# Online Fair Division with Additional Information

Tzeh Yuan Neoh [1] [*]   Jannik Peters [2] [*]   Nicholas Teh [3] [*]

## Abstract

We study the problem of fairly allocating indivisible goods to agents in an online setting, where goods arrive sequentially and must be allocated irrevocably. Focusing on the popular fairness notions of envy-freeness, proportionality, and maximin share fairness (and their approximate variants), we investigate how access to future information changes what guarantees are achievable. Without any information, we prove strong impossibility results even for approximate fairness. With normalization information (agents' total values), we provide an algorithm that achieves stronger fairness guarantees than previously known results, and show matching impossibilities for stronger notions. With frequency predictions (value multisets without order), we design a meta-algorithm that lifts a broad class of offline "share-based" guarantees to the online setting, matching the best-known offline bounds. Finally, we provide learning-augmented variants of both models: under noisy totals or noisy frequency predictions, our guarantees are robust and degrade gracefully with the error parameters.

## 1. Introduction

Fair division of indivisible items is a fundamental problem dealing with the allocation of scarce resources to interested agents in a fair manner (Brams & Taylor, 1996; Moulin, 2003). Such problems arise naturally in many modern ML-driven systems; for instance, platforms must route discrete opportunities (e.g., ad impressions, recommendation exposure, moderation tickets, compute slots) to competing stakeholders, often under operational constraints and fairness requirements. Crucially, these decisions are *sequential*: items arrive over time and must be assigned immediately.

This online nature is central in applications such as multi-resource allocation for cloud computing and shared ML infrastructure (Bei et al., 2022), assigning donations to charities and food banks (Aleksandrov et al., 2015), allocating impressions or exposure in recommendation and marketplace settings (Murray et al., 2020), and distributing content moderation workload over time (Allouah et al., 2023). In such settings, an allocation policy is judged not only by efficiency but also by whether it provides meaningful fairness guarantees to participants.

These applications motivate the study of *online fair division with indivisible goods*.[1] In this setting, we are given a fixed set of agents together with a sequence of indivisible goods arriving sequentially. Upon each arrival, the algorithm observes the agents' values for the current good and must allocate it irrevocably. The goal is that the final allocation is *fair*. Our focus is on standard axioms for indivisible goods—envy-freeness relaxations (EF1/EFX), proportionality relaxations (PROP1), and maximin share (MMS)—and their approximations.

A challenge in the online setting is uncertainty about future arrivals. Without any information about upcoming goods, strong impossibility results are known even in very small instances, and classic offline guarantees can collapse (He et al., 2019; Zhou et al., 2023). At the same time, many ML systems do have partial forecasts: historical data can predict aggregate demand, category frequencies, or total volume over a horizon. This suggests a natural question: which kinds of future information are sufficient to recover meaningful fairness guarantees for indivisible goods?

In this paper, we systematically study how two increasingly informative signals about the future change what is achievable. First, we consider *normalization information* (equivalently, information about each agent's total value over all goods), a common assumption in online fair division (Barman et al., 2022; Gkatzelis et al., 2021; Zhou et al., 2023). This, in spirit, is related to the field of *online algorithms with advice* (refer to a survey by Boyar et al. (2017)). For instance, our problem has a close relation to

---

[*]Alphabetical author ordering. [1]Harvard University, USA [2]Shanghai University of Finance and Economics, China [3]University of Oxford, UK. Correspondence to: Nicholas Teh <nicholas.teh@cs.ox.ac.uk>.

*Proceedings of the 43rd International Conference on Machine Learning*, Seoul, South Korea. PMLR 306, 2026. Copyright 2026 by the author(s).

---

[1]We focus on items with nonnegative valuations, which is a major foundational focus in the fair division literature. While many of our results would extend to chores (i.e., items with nonpositive valuations), we restrict our attention to goods to maintain narrative clarity and focus.

the classical *semi-online scheduling* problem whereby heterogeneous jobs arrive in an online manner and need to be scheduled on homogeneous machines with the goal to minimize the maximum load on any machine (refer to a survey by Dwibedy & Mohanty (2022)). In this setting, it is typically assumed that some additional *offline* (future) information is known about the instance. Examples of such information include the total processing time of all jobs (Albers & Hellwig, 2012; Kellerer et al., 2015), the ratio between the smallest and largest processing time of any job (He & Zhang, 1999), or that the jobs arrive in decreasing order of processing time (Seiden et al., 2000). Second, we study *frequency predictions*, where the algorithm is given, for each agent, the multiset of values that will appear, but not the arrival order. This type of partial information has also been studied in learning-augmented and semi-online models (e.g., knapsack/bin packing/matching) (Mehta et al., 2007; Im et al., 2021; Angelopoulos et al., 2023; Balseiro et al., 2023). We complement both models with *learning-augmented* variants, where the advice may be noisy.

## 1.1. Our Contributions

We study online fair division of indivisible goods under an *adaptive* adversary and *ex post* evaluation of the final allocation.

We begin in Section 3 by considering the online fair division model *without* additional information about future goods. We show that for $n \geq 2$, no deterministic online algorithm can guarantee any positive approximation of EF1. We also use a lower bound of Benadè et al. (2018) to show that, even for two agents with bounded item values, any deterministic online algorithm can be forced to output an allocation that violates PROP1 on instances with sufficiently many goods (even with bounded item values); consequently, no algorithm can guarantee PROP1 in this fully online model.[2]

Motivated by prior work that assumes normalized valuations (or equivalently provides each agent's total value in advance) (Gkatzelis et al., 2021; Barman et al., 2022; Zhou et al., 2023), Section 4 develops a new algorithm that guarantees: (i) PROP1 for any number of agents, and (ii) EF1 (and thus $1/2$-MMS) simultaneously for two agents. This does not follow directly from the semi-online MMS result of Zhou et al. (2023): their rule can violate PROP1 (and therefore EF1) even for $n = 2$. Our algorithm uses a different removal rule, based on the largest good outside an agent's bundle, which is exactly the type of one-good condition appearing in PROP1 and EF1. We complement these guarantees with matching impossibility results: even with normalization information, no positive approximation to EFX is possible already for $n = 2$, and no positive approxi-

mation to EF1 is possible for $n \geq 3$. These results clarify what normalization information is sufficient for in online fair division. In Section 4.1, we allow the predicted total values to be inaccurate and show how the guarantees change as a function of the prediction error.

We prove robustness guarantees for the same algorithm: an additive approximate EF1 bound for $n = 2$, and a multiplicative $\kappa$-PROP1 guarantee for general $n$, parameterized by the advice accuracy. This explicitly connects online fair division to learning-augmented analysis, while preserving ex post guarantees.

To motivate our investigation into whether stronger fairness guarantees can be achieved by providing the algorithm with richer information, in Section 5, we consider a model where the online algorithm has access to *frequency predictions* (i.e., for each agent and value, a predictor tells us the frequency with which this value will appear among the agents' valuations for items that will arrive). This is motivated by similar notions in the online knapsack (Im et al., 2021), online bin packing (Angelopoulos et al., 2023), and online matching (Mehta et al., 2007) literature. Here, we prove a general result: any *share-based* fairness notion that can be achieved by an offline algorithm in the single-shot setting[3] can also be achieved by an online algorithm with frequency predictions—we give an online algorithm that achieves the same share guarantee. In particular, this implies that the currently best known $\left(\frac{3}{4} + \frac{3}{3836}\right)$-MMS approximation guarantee can also be obtained in our setting using frequency predictions for any number of agents, thereby improving on the (tight) $0.5$-MMS guarantee under normalized valuations established by Zhou et al. (2023). We also show that that EFX can be achieved for two agents, and complement this by showing that for three or more agents, no positive approximation to EFX is possible. In Section 5.1, we provide a learning-augmented variant: when the predicted multisets are noisy, the guarantee degrades smoothly with an explicit instantiation error parameter.

Finally, motivated by the relation of our problem to online scheduling (where many works consider the setting with identical machines (Dwibedy & Mohanty, 2022)), we study the setting where agents have identical valuation functions (i.e., each good is valued the same by every agent). In this setting, an algorithm by Elkind et al. (2025a) implies that EF1 can be achieved without any information about the future. We complement this by showing that any positive EFX approximation remains impossible for (i) two agents without any information, and (ii) three agents or more given normalization information. For two agents given the normal-

---

[2]We also clarify a flaw in the construction of Kahana & Hazon (2023) that was claimed to establish a similar impossibility.

[3]Some examples studied in the literature include *round robin share* (RRS) (Conitzer et al., 2017; Gourvès et al., 2021), *minimum EFX share* (MXS) (Caragiannis et al., 2023), and $\left(\frac{3}{4} + \frac{3}{3836}\right)$-MMS (Akrami & Garg, 2024).

ization information, we provide an algorithm that returns a $\frac{\sqrt{5}-1}{2}$-EFX allocation, and show that this bound is tight (i.e., no better approximation to EFX is possible).

Finally, although we focus on deterministic algorithms, our impossibility results extend to randomized algorithms under our adaptive adversary and ex-post model (Appendix A).

### 1.2. Further Related Work

**Online Fair Division.**   Earlier work on online fair division was motivated by applications such as food bank allocation (Aleksandrov et al., 2015) and studied fairness and incentives under restricted valuation classes; see Aleksandrov & Walsh (2020) for a survey. More recent papers analyze how fairness evolves over time (Benadè et al., 2018), fairness-efficiency tradeoffs (Zeng & Psomas, 2020), and the impact of changing the past on achievable guarantees (He et al., 2019). Closest to our work, Zhou et al. (2023) studied the problem given normalized valuations, focusing on MMS; we extend the study to envy- and proportionality-based axioms and to richer prediction models. A recent follow-up work by Choo et al. (2026) focuses specifically on approximate PROP1 in online fair division, showing lower bounds for several natural greedy rules under adaptive adversaries, as well as positive guarantees under weaker adversaries and with predictions of the maximum item value.

Recent work also explores structured valuation domains in online fair division. In particular, Amanatidis et al. (2025) study personalized two-value instances, and Wang & Wei (2026) investigate online fair allocations under binary valuations beyond classical food-bank models. These lines of work highlight what becomes possible under strong domain restrictions. In contrast, we focus on *general additive valuations* and ask how different *forms of future information* (normalization information, frequency predictions) enable or preclude classic fairness guarantees.

Elkind et al. (2025a) consider a fully-informed setting where the sequence of incoming items is fixed, and all valuations over future goods is known in advance with the goal being to achieve an approximately envy-free allocation at *every round prefix*, a stronger requirement than ours.

**Learning-Augmented Online Fair Division**   A growing body of work studies online algorithms that exploit predictions about the future and analyzes consistency (when predictions are correct) and robustness (when they are not). Within fair division, much of this literature focuses on *divisible* goods (Banerjee et al., 2022; 2023; Cohen & Panigrahi, 2023; Golrezaei et al., 2023; An et al., 2024), where allocations can be fractionally adjusted and this differs substantially from the indivisible setting. For indivisible goods, existing learning-augmented work often targets objectives other than the classic envy/proportionality relax-

ations: Spaeh & Ene (2023) study online ad allocation with predictions under cardinality constraints with a welfare objective; Balkanski et al. (2023) and Cohen et al. (2024) focus on truthfulness/strategyproofness and makespan-style guarantees (closely related to MMS via scheduling analogies). Our work is complementary in two aspects: (i) we focus on fundamental *fairness axioms* for indivisible goods (EF1/EFX/PROP1/MMS), and (ii) we provide *learning-augmented* guarantees tailored to these axioms under both noisy normalization information and frequency predictions.

**Partial Information and Bandit Feedback.**   Several models study online fair division when valuation information is incomplete or noisy. Benadè et al. (2022) consider distributional valuations with only partial (ordinal) access, and show high-probability fairness under shared distributions. A recent line of work connects online fair division to bandit learning and regret minimization, where valuation feedback is noisy or contextual (Procaccia et al., 2024; Yamada et al., 2024; Verma et al., 2025; Schiffer & Zhang, 2025). These papers typically study uncertainty about values or distributions, observe limited or noisy feedback, and evaluate performance through regret, expectation, or fairness over time. Our setting is different: when a good arrives, we observe the full valuation vector, the adversary may choose the arrival order, and the guarantee is ex post fairness of the final allocation. Thus, our question is not how quickly values can be learned, but which predictions about future goods allow one to obtain offline-style guarantees under irrevocable allocation.

**Online Social Choice**   Besides fair division, online algorithms and related settings have been studied for various social choice topics. For instance, in the multiwinner voting literature, various papers have studied the online (or temporal selection) of candidates (Lackner, 2020; Alouf-Heffetz et al., 2022; Do et al., 2022; Elkind et al., 2022; Lackner & Maly, 2023; Chandak et al., 2024; Elkind et al., 2024a;b; Zech et al., 2024; Elkind et al., 2025b;c; Phillips et al., 2026; Teh, 2026). Bullinger & Romen (2023; 2024) studied online coalition formation in which agents arrive over time and need to be assigned to coalitions. Further, there are various works studying online fair matching problems in which either the matching participants arrive over time (Esmaeili et al., 2023; Hosseini et al., 2024) or the preferences of the participants need to be discovered in an online process (Hosseini et al., 2021; Peters, 2022; Amanatidis et al., 2024).

**Relation to prophet inequalities.**   Our model is closer to semi-online/advice and learning-augmented online algorithms than to prophet inequalities. Prophet inequality models are typically stochastic optimal-stopping or online-selection problems benchmarked against a prophet who sees

| Fairness Property | No Information | Normalization Information | Frequency Predictions |
|---|---|---|---|
| EFX | ✗† | ✗† | ✓ ($n = 2$); ✗† ($n \geq 3$) |
| EF1 | ✗† | ✓ ($n = 2$); ✗† ($n \geq 3$) | ✓ ($n = 2$); ? ($n \geq 3$) |
| PROP1 | ✗ | ✓ | ✓ |
| MMS | ✗† (Zhou et al., 2023) | $1/2$ ($n = 2$); ✗† ($n \geq 3$) (Zhou et al., 2023) | $\frac{3}{4} + \frac{3}{3836}$ |

*Table 1.* Overview of fairness guarantees across information models. ✓ denotes exact feasibility, numeric entries denote a multiplicative approximation factor, ✗ denotes impossibility of exact satisfaction, and ✗† denotes that no strictly positive competitive ratio is achievable. For PROP1 without information, we rule out exact satisfaction for instances with sufficiently many goods, but do not rule out positive approximation factors.

the realizations in advance (Correa et al., 2019). In contrast, our setting is adversarial and multi-agent: every good must be allocated irrevocably, and performance is evaluated ex post through fairness of the final allocation rather than through expected reward from selecting a subset of arrivals.

## 2. Preliminaries

For each positive integer $z$, we denote $[z] := \{1, \ldots, z\}$. Throughout the paper, we assume we are given a set $N = [n]$ of (fixed) *agents* and a set $G = \{g_1, \ldots, g_m\}$ of $m$ *indivisible goods* arriving online. We label the goods $g_1, \ldots, g_m$ in the order that they arrive. Each agent $i \in N$ has a non-negative *valuation function* $v_i \colon 2^G \to \mathbb{R}_{\geq 0}$. We write $v$ instead of $v_i$ when all agents have identical valuation functions. As with most works in the fair division literature, we assume that valuation functions are *additive*, i.e., for any subset of goods $S \subseteq G$, $v_i(S) = \sum_{g \in S} v_i(\{g\})$. For convenience, we write $v_i(g)$ instead of $v_i(\{g\})$ for a single good $g$. Every subset of goods in $G$ can also be referred to as a *bundle*. An *allocation* $\mathcal{A} = (A_1, \ldots, A_n)$ is a *partition* of the goods into bundles, with agent $i \in N$ receiving bundle $A_i$.

One of the most common fairness notions considered in fair division is *envy-freeness* (EF): every agent's value for their own bundle should be at least as much as their value for any other agent's bundle. However, the existence of EF allocations cannot be guaranteed, even in the simplest case of two agents and a single good. Thus, we consider the two most common relaxations of EF. The first is a relatively strong and widely studied EF relaxation: *envy-freeness up to any good*.

**Definition 2.1** (EFX). An allocation $\mathcal{A} = (A_1, \ldots, A_n)$ is *envy-free up to any good (EFX)* if for every pair of agents $i, j \in N$ and every $g \in A_j$, we have that $v_i(A_i) \geq v_i(A_j \setminus \{g\})$.

In the offline setting, the existence of EFX allocations for more than three agents is still an open problem, and remains one of "fair division's most enigmatic question(s)" (Procaccia, 2020). As such, many works focus on a relatively weaker but still natural variant of envy-freeness:

*envy-freeness up to one good.*

**Definition 2.2** (EF1). An allocation $\mathcal{A} = (A_1, \ldots, A_n)$ is *envy-free up to one good (EF1)* if for every pair of agents $i, j \in N$ with $A_j \neq \varnothing$ there exists a $g \in A_j$ such that $v_i(A_i) \geq v_i(A_j \setminus \{g\})$.

It is easy to see that EFX implies EF1. Furthermore, there are several algorithms that produce an EF1 allocation, such as the envy-cycle elimination algorithm (Lipton et al., 2004) or the classic round-robin procedure.

We also consider a weaker fairness property, called *proportionality* (PROP), which requires that each agent receives her "proportional share"—that is, at least a $1/n$-th fraction of the total value of all goods according to her own valuation (Steinhaus, 1948). Numerous prior works have looked into this fairness concept and its relaxed variants in the offline setting (Amanatidis et al., 2023; Aziz et al., 2023). Similar to the case of EF, a proportional allocation is not always guaranteed to exist, even with just two agents and a single good. We focus on the analogous "up to one good" relaxation commonly considered in the literature.[4]

**Definition 2.3** (PROP1). An allocation $\mathcal{A} = (A_1, \ldots, A_n)$ is *proportional up to one good (PROP1)* if for every agent $i \in N$, either $A_i = G$ or there exists a $g \in G \setminus A_i$ such that $v_i(A_i \cup \{g\}) \geq \frac{v_i(G)}{n}$.

It is easy to see that EF1 implies PROP1. Moreover, while EF and PROP are equivalent in the case of $n = 2$, the same relationship cannot be established for EFX, EF1, or PROP1.

The last fairness notion we consider in this work is *maximin share fairness* (MMS), which was also the focus of Zhou et al. (2023). Intuitively, MMS guarantees that each agent receives a bundle she values at least as much as she would have gotten if she were allowed to partition all goods into $n$ bundles and then receive the least valuable bundle (according to her own valuation).

**Definition 2.4** (MMS). Let $\Pi(G)$ be the set of all $n$-partitions of $G$. The *maximin share* of each agent $i \in N$ is

---

[4]A stronger variant of PROP1 would be proportionality *up to any good* (PROPX). However, it is not always satisfiable even in the single-shot setting (Aziz et al., 2020).

defined as $MMS_i := \max_{\mathcal{X} \in \Pi(G)} \min_{j \in N} \{v_i(X_j)\}$. Then, an allocation $\mathcal{A} = (A_1, \ldots, A_n)$ is maximin share fair (MMS) if $v_i(A_i) \geq MMS_i$ for all $i \in N$.

While PROP implies MMS, MMS implies PROP1 (Caragiannis et al., 2023). Finally, we also study approximate versions of the properties defined above.[5] For $\alpha \in [0, 1]$, we say that an allocation $\mathcal{A}$ is:

- $\alpha$-EFX if for every $i, j \in N$, either $A_j = \varnothing$ or for every $g \in A_j$, we have that $v_i(A_i) \geq \alpha \cdot v_i(A_j \setminus \{g\})$;

- $\alpha$-EF1 if for every $i, j \in N$, either $A_j = \varnothing$ or there exists a $g \in A_j$ such that $v_i(A_i) \geq \alpha \cdot v_i(A_j \setminus \{g\})$;

- $\alpha$-PROP1 if for every $i \in N$, either $A_i = G$ or there exists a $g \in G \setminus A_i$ such that $v_i(A_i \cup \{g\}) \geq \alpha \cdot \frac{v_i(G)}{n}$;

- $\alpha$-MMS if for every $i \in N$, $v_i(A_i) \geq \alpha \cdot MMS_i$.

Note that for any $\alpha \in [0, 1]$, $\alpha$-EFX implies $\alpha$-EF1, and for any $n$, EF1 implies $\frac{1}{n}$-MMS (Amanatidis et al., 2018; Segal-Halevi & Suksompong, 2019).

We equivalently say that a rule satisfies the property if any allocation returned by the rule on a problem instance satisfies the property. Note that when $\alpha = 1$, the property is satisfied exactly and the lower $\alpha \in [0, 1]$ is, the worse the approximation gets.

**Online Setting**  As is standard in online fair division literature, we assume an adversarial model where an adversary both constructs the instance and controls the sequence of arriving goods. As we consider deterministic online algorithms (again, another standard consideration), the adversary may be adaptive, choosing each item's valuation profile based on the algorithm's prior allocation decisions. The algorithm is given the number of agents $n$, along with some information (if any), which depends on the setting we are considering. When a good $g$ arrives, the algorithm observes all $\{v_i(g)\}_{i \in N}$ and must immediately and irrevocably assign $g$ to an agent.[6] We evaluate fairness *ex post* on the realized final allocation. We then measure performance of the algorithm by the *competitive ratio*, i.e., the worst-case approximation factor of the algorithm (with respect to the fairness objective) of the final allocation $\mathcal{A}$ over all online instances.

---

[5]We consider *multiplicative* approximations of these fairness properties, which is a well-established and common approach in fair division (as compared to *additive* approximations).

[6]This corresponds to the notion of *completeness*, which is also a standard assumption in almost all prior work on online fair division, and most prior work in the classic fair division model. An exception in the latter is the setting involving the allocation of goods *with charity*, where an objective is to minimize the number of unallocated goods (or those given to charity), on top of ensuring an approximately fair allocation.

## 3. Impossibilities Without Information

We begin by considering the basic setting of online fair division *without* any information—that is, goods arrive one at a time and must be irrevocably allocated to agents, with no knowledge of future items. Zhou et al. (2023, Theorem A.1) proved that in the absence of any information, there does not exist any online algorithm with a competitive ratio strictly larger than $0$ with respect to MMS, even for $n = 2$. Here, we consider (the approximate variants of) two other commonly studied fairness concepts in the fair division literature (EF and PROP) in the same setting.

We begin with the envy-freeness relaxations. Benadè et al. (2018) showed that no online algorithm, in the absence of information, can achieve EF1 (and hence EFX). We strengthen this by proving a stronger impossibility result: no online algorithm in this setting can guarantee *any* positive approximation to EF1.

**Proposition 3.1.** *For $n \geq 2$, without future information, there does not exist any deterministic online algorithm with a competitive ratio strictly larger than $0$ with respect to* EF1.

Given this, a natural follow-up question is whether it is possible to approximate a weaker fairness notion in this setting. Prior work by Kahana & Hazon (2023, Thm. 13) claims that no algorithm can satisfy PROP1 in this setting. We show that their example used to disprove the existence of an algorithm that returns a PROP1 allocation is incorrect. *Example* 3.2 (Kahana & Hazon (2023)). Consider an instance with $n = 2$ agents and let goods arrive in the order $g_1, \ldots, g_m$. Also consider some round $t$ where $g_t$ arrives. If $t$ is odd, then let $v_1(g_t) = v_2(g_t) = 1$. If $t$ is even and the good $g_{t-1}$ was allocated to agent 1, then let $v_1(g_t) = 1$ and $v_2(g_t) = 0.875$; symmetrically, if $g_{t-1}$ was allocated to agent 2, then let $v_1(g_t) = 0.875$ and $v_2(g_t) = 1$.

Kahana & Hazon (2023) claimed that no online algorithm can achieve PROP1 in the above instance. However, observe that a simple online algorithm that always assigns two goods to the first agent, then two goods to the second agent, and so on, in an alternating fashion would satisfy PROP1. Their proof fails in the induction, in which the base case (for $t = 0$) is not true. Nevertheless, we are able to reason about the non-existence of a PROP1 allocation from the following result by Benadè et al. (2018).

**Theorem 3.3** (Benadè et al. (2018))**.** *There exist constants $c > 0$ and $m_0 \in \mathbb{N}$ such that for every integer $m \geq m_0$ and every deterministic online allocation algorithm, there exists an instance with two agents and $m$ goods satisfying $v_i(g) \in [0, 1]$ for all agents $i$ and goods $g$, such that if the algorithm outputs allocation $\mathcal{A}$, then $\max\{v_1(A_2) - v_1(A_1), v_2(A_1) - v_2(A_2)\} \geq c\sqrt{m}$.*

Using the above result, we can strengthen the conclusion

to an explicit online impossibility for PROP1 (even under bounded item values): for instances with sufficiently many goods, an adaptive adversary can force any deterministic online algorithm to output an allocation that violates PROP1.

**Corollary 3.4.** *There exists $m_1 \in \mathbb{N}$ such that for every $m \geq m_1$ and every deterministic online allocation algorithm, there is an instance with two agents and $m$ goods satisfying $v_i(g) \in [0, 1]$ for all $i$ and $g$ on which the algorithm's final allocation is not PROP1.*

*Proof.* By Theorem 3.3, there exist constants $c > 0$ and $m_0$ such that for every $m \geq m_0$ and every deterministic online algorithm, an adaptive adversary can generate an instance with $v_i(g) \in [0, 1]$ for which the algorithm outputs an allocation $A = (A_1, A_2)$ satisfying

$$\max\{v_1(A_2) - v_1(A_1), \ v_2(A_1) - v_2(A_2)\} \ \geq \ c\sqrt{m}.$$

Choose $m_1$ so that $m_1 \geq m_0$ and $c\sqrt{m_1} > 2$. Then for any $m \geq m_1$, the produced instance satisfies that for at least one agent (say agent 1) we have $v_1(A_2) - v_1(A_1) > 2$. Since $v_1(g) \leq 1$ for every good, for any $g \in A_2$ we have $v_1(A_1 \cup \{g\}) \leq v_1(A_1) + 1 < \frac{1}{2}(v_1(A_1) + v_1(A_2)) = v_1(G)/2$, so agent 1 violates PROP1. □

## 4. Normalization Information

In the previous section, we observed that even a weak (but commonly studied) fairness notion—PROP1— cannot be guaranteed without any information about future goods. This naturally raises the question: if we are given *some* information about future goods, can we obtain better approximation guarantees, or perhaps even satisfy certain fairness properties in the online setting?

Here, we begin to address this question by considering a very common kind of information studied in the literature (both in the online fair division setting, and in other contexts): *normalization* information. For this, we assume that the agents' valuations over all goods sum to 1 (in our setting, this is, without loss of generality, equivalent to knowing the sum of valuations for each agent over all goods). This assumption is fairly common in the online fair division literature, and was studied by Zhou et al. (2023) for MMS, Banerjee et al. (2022) and Gkatzelis et al. (2021) in the divisible goods setting, and Banerjee et al. (2023) in the public goods setting.

In our setting, Zhou et al. (2023) provided an algorithm for computing a 0.5-MMS allocation for $n = 2$, but showed that no online algorithm can have a positive competitive ratio for MMS for $n \geq 3$. Here, we generalize and modify the algorithm of Zhou et al. (2023, Algorithm 1) to obtain an algorithm that returns an allocation that satisfies EF1 for two agents (and hence 0.5-MMS for two agents), and

PROP1 for any number of agents.[7] We further note that the algorithm of Zhou et al. (2023) does not satisfy PROP1 (and therefore also not EF1) even for the case of two agents.[8]

The key difference from the approximate MMS algorithm of Zhou et al. (2023) is the rule used to remove agents from further competition. For each active agent $i$, Algorithm 1 keeps track of the largest value of a good currently outside $i$'s bundle, denoted $M_i$. When good $g_t$ arrives, the algorithm computes $x_i = v_i(A_i) + \frac{n-1}{n} \max\{M_i, v_i(g_t)\}$. If $x_i \geq 1/n$, then the proof shows that, in the final allocation, agent $i$'s bundle together with one good outside it has value at least $1/n$. This is the PROP1 condition. The coefficient $(n - 1)/n$ is chosen so that the summation argument in the proof works for agents who remain active until the end. When $n = 2$, the same inequality also implies EF1. Thus the difference from Zhou et al. (2023) is not just the use of active agents, but the use of the largest outside good and the threshold $1/n$, which are tailored to PROP1 and EF1.

For each agent $i$, maintain a bundle $A_i$ and a status set of active agents $N'$. For each $t \in \{0, 1, \ldots, m\}$, let $A_i^t$ denote the bundle held by agent $i$ after the first $t$ goods $g_1, \ldots, g_t$ have been allocated (so $A_i^0 = \varnothing$ and $A_i^m = A_i$). Define the outside-max of agent $i$ after $t$ allocations as $M_i^t := \max\{v_i(g) : g \in \bigcup_{j \neq i} A_j^t\}$. Thus, at iteration $t$ (when $g_t$ arrives), Algorithm 1 uses the current bundles $A_i^{t-1}$ and the outside-max $M_i^{t-1}$.

Note that in Algorithm 1, $A_i$ inside the loop denotes the bundle held by agent $i$ *before* allocating $g_t$.

We first show that Algorithm 1 returns an EF1 allocation in the case of two agents.

**Theorem 4.1.** *For $n = 2$, given normalization information, Algorithm 1 returns an* EF1 *allocation.*

Since EF1 implies 0.5-MMS when there are two agents (Amanatidis et al., 2018; Segal-Halevi & Suksompong, 2019), we also obtain the same 0.5-MMS guarantee as Zhou et al. (2023, Algorithm 1). Next, we show that the algorithm returns a PROP1 allocation for any number of agents, thereby also providing the first satisfiable fairness property in this setting for three (or more) agents.

**Theorem 4.2.** *For $n \geq 2$, given normalization information, Algorithm 1 returns a* PROP1 *allocation.*

Note that Algorithm 1 can be implemented in $\mathcal{O}(mn)$ time. Each round computes $x_i$ for active agents, selects the max-

---

[7]Note that PROP1 is strictly weaker than (and hence not equivalent to) EF1, even when $n = 2$ (Aziz et al., 2023).

[8]To see this, consider an instance with six goods and two agents with identical valuations. Let the agents value the goods at $1/4 - \varepsilon, \varepsilon, 3/16, 3/16, 3/16, 3/16$ (in order of arrival). The algorithm of Zhou et al. (2023) could assign the first and second good to the first agent and the last four goods to the second agent leading to a PROP1 violation of the first agent, as $1/4 + 3/16 < 1/2$.

---

**Algorithm 1** EF1 for $n = 2$ and PROP1 for $n \geq 2$ given normalization information

1: Initialize $A_i \leftarrow \varnothing$ and $M_i \leftarrow 0$ for all $i \in N$, and $N' \leftarrow N$
2: **for** $t = 1$ **to** $m$ **do**
3:     **if** $N' = \varnothing$ **then**
4:         Assign all remaining goods $\{g_t, \ldots, g_m\}$ to agent 1 and **return** $\mathcal{A} = (A_1, \ldots, A_n)$
5:     **end if**
6:     **for** each $i \in N'$ **do**
7:         $x_i \leftarrow v_i(A_i) + \frac{n-1}{n} \cdot \max\{M_i, v_i(g_t)\}$
8:         **if** $x_i \geq \frac{1}{n}$ **then**
9:             $N' \leftarrow N' \setminus \{i\}$
10:         **end if**
11:     **end for**
12:     **if** $N' = \varnothing$ **then**
13:         Assign all remaining goods $\{g_t, \ldots, g_m\}$ to agent 1 and **return** $\mathcal{A} = (A_1, \ldots, A_n)$
14:     **end if**
15:     Choose $k \in \arg\max_{j \in N'} v_j(g_t)$, breaking ties by smallest $x_j$
16:     $A_k \leftarrow A_k \cup \{g_t\}$
17:     **for all** $i \in N \setminus \{k\}$ **do**
18:         $M_i \leftarrow \max\{M_i, v_i(g_t)\}$
19:     **end for**
20: **end for**
21: **return** $\mathcal{A} = (A_1, \ldots, A_n)$.

---

imizing agent, and updates the outside-max values $M_i$ for all $i \neq k$ in $\mathcal{O}(n)$ time.

We complement these positive results with two impossibility results, demonstrating that no algorithm can achieve better guarantees for the considered fairness notions in this setting with normalized information. We first consider the stronger fairness notion of EFX and show that no online algorithm can guarantee a positive competitive ratio even for two agents.

**Theorem 4.3.** *For $n \geq 2$, given normalization information, there does not exist any online algorithm with a competitive ratio strictly larger than $0$ with respect to* EFX.

Next, we turn to EF1 and show that for three or more agents, no online algorithm can guarantee a positive competitive ratio, thereby complementing our positive result for $n = 2$.

**Theorem 4.4.** *For $n \geq 3$, given normalization information, there does not exist any online algorithm with a competitive ratio strictly larger than $0$ with respect to* EF1.

### 4.1. Learning-Augmented Guarantees under Noisy Normalization Information

In many applications the normalization constants $T_i := v_i(G)$ are obtained from prediction and may be inaccurate.

We model this by assuming the algorithm is given bounds $\underline{T}_i$ and $\overline{T}_i$ such that $0 < \underline{T}_i \leq T_i \leq \overline{T}_i$. Define the additive uncertainty $\rho_i := \overline{T}_i - \underline{T}_i$ and the multiplicative accuracy $\kappa_i := \underline{T}_i/\overline{T}_i \in (0, 1]$. We run Algorithm 1 on the scaled valuations $\tilde{v}_i(S) := v_i(S)/\underline{T}_i$ for all $S \subseteq G$. Then, our result is as follows.

**Theorem 4.5.** *Let $\mathcal{A}$ be the allocation returned by Algorithm 1 when run on the scaled valuations $\widetilde{v}_i := v_i/\underline{T}_i$.*

  *(i) If $n = 2$, for every $i, j \in N$ with $A_j \neq \varnothing$, there exists a good $g \in A_j$ such that $v_i(A_i) \geq v_i(A_j \setminus \{g\}) - \rho_i$.*

  *(ii) For any $n \geq 2$ and every $i \in N$, either $A_i = G$ or there exists a good $g \in G \setminus A_i$ such that $v_i(A_i \cup \{g\}) \geq \kappa_i \cdot \frac{v_i(G)}{n}$. In particular, letting $\kappa := \min_{i \in N} \kappa_i$, the allocation is $\kappa$-PROP1.*

When the advice is exact (i.e., $\underline{T}_i = \overline{T}_i = T_i$ for all $i$), (i) reduces to Theorem 4.1 and (ii) reduces to Theorem 4.2. With inaccurate advice, the EF1 guarantee in (i) loses at most $\rho_i$, while the PROP1 guarantee in (ii) keeps a multiplicative factor of $\kappa_i = \underline{T}_i/\overline{T}_i$.

Unlike PROP1, robustness for EF1 under noisy normalization information is naturally additive: in this advice model, no multiplicative $\alpha$-EF1 guarantee (for any $\alpha > 0$) is possible in general; see Appendix C.1.

## 5. Frequency Predictions

We next consider a stronger form of information, called *frequency predictions*, inspired by similar notions studied in the online knapsack (Im et al., 2021), online bin packing (Angelopoulos et al., 2023), and online matching (Mehta et al., 2007) literature. These types of predictive information have also been explored in the context of revenue management (Balseiro et al., 2023). Specifically, we assume access to a predictor that, for each agent and value, tells us the frequency with which this value will appear among the agent's valuations for goods that will arrive. Formally, for each agent $i \in N$, $V_i$ denotes the multiset containing values that agent $i$ will have for goods that will arrive (we call this the *frequency multiset*). At a high level, $V_i$ captures *distributional* information about agent $i$'s future utilities while discarding the arrival order. We emphasize that our guarantees are worst-case with respect to an *adversarial permutation* of the multiset: the predictor reveals which values will appear and how many times, but not *when* they appear nor which concrete item carries which value. This is directly analogous to semi-online models in scheduling and packing, where knowing the multiset of future sizes but not their order is a standard form of partial information.

From a modeling perspective, frequency predictions are particularly natural in settings where goods fall into a mod-

erate number of *types* (categories). If each arriving good has a type $\tau(g) \in \mathcal{T}$ and agent values are (approximately) type-dependent, then forecasting the number of arrivals of each type is a standard prediction task. Given estimated type counts and known (or learned) type utilities $(u_i)_{i \in N}$, one obtains an induced frequency multiset for each agent without predicting the full arrival sequence.

We are interested in understanding whether access to frequency predictions can improve the fairness guarantees in the online setting. To this end, we establish a considerably general result: any *share-based* fairness notion (Babaioff & Feige, 2024) that is achievable in the single-shot setting can also be obtained via an online algorithm with frequency predictions. Formally, a share $s$ is a function mapping each frequency multiset $V_i$ and number of agents $n$ to a nonnegative share value $s(V_i, n) \in \mathbb{R}_{\geq 0}$. A share is *feasible*, if for every instance, there exists an allocation giving each agent a utility of at least $s(V_i, n)$. Examples of feasible shares (i.e., shown to be achievable in the single-shot setting) include *round robin share* (RRS) (Conitzer et al., 2017; Gourvès et al., 2021), *minimum EFX share* (MXS) (Caragiannis et al., 2023), or $\left(\frac{3}{4} + \frac{3}{3836}\right)$-MMS (Akrami & Garg, 2024). Then, our result is as follows.

For $t \in [m]$ and agent $i$, let $V_i^t$ be the multiset of $i$'s values for the remaining goods $\{g_t, \ldots, g_m\}$. The corresponding ordered, or identical ordering (IDO), valuation $v_i^{t,\mathsf{IDO}}$ is obtained by sorting $V_i^t$ in non-increasing order and assigning the $k$-th largest value to good $g_{t+k-1}$. Thus, in the valuation profile $v^{t,\mathsf{IDO}} = (v_1^{t,\mathsf{IDO}}, \ldots, v_n^{t,\mathsf{IDO}})$, all agents weakly rank the remaining goods in the common order $g_t, g_{t+1}, \ldots, g_m$.

**Theorem 5.1.** *Let $s$ be a feasible share in the single-shot fair division setting. Then, in the online fair division setting, there exists an online algorithm using frequency predictions that guarantees each agent a bundle value of at least $s(V_i, n)$.*

The proof of Theorem 5.1 is constructive. First, compute an offline allocation that achieves the share $s$ for the ordered instance induced by the frequency multisets. Since all agents rank the goods in the same order in this instance, this allocation determines a picking sequence $\pi^1$. Algorithm 2 then uses this picking sequence online. At time $t$, it keeps the true values of the current good $g_t$, fills the remaining goods according to the ordered residual multisets, simulates the remaining picking sequence, and assigns $g_t$ to the agent who would pick it in that simulation.

*Example* 5.2. Suppose $n = 2$, $m = 3$, $V_1 = \{7, 4, 1\}$, and $V_2 = \{6, 5, 1\}$. The IDO instance on goods $g_1, g_2, g_3$ has $v_1^{1,\mathsf{IDO}}(g_1, g_2, g_3) = (7, 4, 1)$ and $v_2^{1,\mathsf{IDO}}(g_1, g_2, g_3) = (6, 5, 1)$, so both agents rank $g_1$ above $g_2$ above $g_3$. If an offline share-achieving allocation on this ordered instance is $X_1 = \{g_1, g_3\}$ and $X_2 = \{g_2\}$, then the induced picking sequence is $\pi^1 = (1, 2, 1)$. Algorithm 2 does not assume

that the actual arrival sequence is ordered. After each arrival, it removes the realized values from the residual multisets and runs the remaining picking sequence on the updated valuation profile.

Thus, whenever the offline share $s$ is achievable in polynomial time, the induced online algorithm is also polynomial time (with per-round overhead polynomial in $n$).

As discussed before, with normalization information, no online algorithm can achieve better than $0.5$-MMS for $n = 2$ or even any positive approximation to MMS for $n > 2$. In contrast, frequency predictions allow us to leverage the best-known MMS approximation in the single-shot setting for any number of agents, giving us the following result.

**Corollary 5.3.** *For $n \geq 2$, given frequency predictions, there exists an online algorithm that returns a $\left(\frac{3}{4} + \frac{3}{3836}\right)$-MMS allocation.*

Interestingly, we can further extend this framework to show that EFX is achievable for two agents with frequency predictions, by leveraging the leximin++ cut-and-choose procedure of Plaut & Roughgarden (2018, Algorithms 1 and 2). Our result is as follows.

**Theorem 5.4.** *For $n = 2$, given frequency predictions, there exists an online algorithm that always returns an EFX allocation.*

Theorem 5.4 does not settle EF1 for $n \geq 3$. The theorem applies to share guarantees: for each agent $i$, the target is a number depending only on $(V_i, n)$, and the conclusion is a lower bound on $v_i(A_i)$. EF1 is different, it compares agent $i$'s bundle with each other agent's final bundle after the removal of one good. This pairwise comparison is not captured by the share guarantees used in Theorem 5.4.

Together with the results of Section 4, this raises a natural follow-up question: does an EF1 allocation exist for $n \geq 3$ given frequency predictions? Despite considerable effort, this question remains elusive and unresolved. We conjecture that no online algorithm can guarantee EF1 in this setting, even with frequency predictions, though achieving a positive competitive ratio may still be possible. We leave this as an open problem. To end this section, we complement the positive result of Theorem 5.4 by showing a strong impossibility result with respect to EFX for any higher values of $n$, even with frequency predictions.

**Theorem 5.5.** *For $n \geq 3$, given frequency predictions, there does not exist any online algorithm with a competitive ratio strictly larger than $0$ with respect to EFX.*

These negative results highlight a significant gap that remains between the offline (single-shot) and online settings: in the former, while an exact EFX allocation is guaranteed to exist for $n = 3$ (Chaudhury et al., 2024) and a $0.618$-EFX allocation exists in general for all $n$ (Amanatidis et al.,

2020), no comparable guarantees hold in the online setting, even with frequency predictions.

## 5.1. Learning-Augmented Guarantees under Noisy Frequency Predictions

We now consider a learning-augmented variant of frequency predictions: instead of the true frequency multisets $(V_i)_{i \in N}$, the algorithm receives predicted multisets $(\widehat{V}_i)_{i \in N}$. Since each $\widehat{V}_i$ is an unordered multiset, an online algorithm must *instantiate* $\widehat{V}_i$ online against the realized arrival order, which induces a virtual valuation $\widetilde{v}_i$ and an agent-specific relative error $\varepsilon_i$ (formalized in Appendix D.1). Our share guarantee degrades gracefully as a function of this error.

**Theorem 5.6.** *Let $s$ be a feasible share. There exists an online algorithm that, given the predicted multisets $(\widehat{V}_i)_{i \in N}$, outputs an allocation $\mathcal{A}$ such that for every agent $i \in N$, $v_i(A_i) \geq (1 - \varepsilon_i) \cdot s(\widehat{V}_i, n)$, where $\varepsilon_i \in [0, 1]$ is the relative instantiation error induced by the algorithm's online instantiation rule.*

The above result admits the standard learning-augmented algorithm guarantees of consistency, robustness, and smoothness. If the predictions are perfect, i.e., $\widehat{V}_i = V_i$ for all $i \in N$, then instantiating online by always choosing $\widetilde{v}_i(g_t) = v_i(g_t)$ gives us $\eta_i = 0$ and thus, $\varepsilon_i = 0$, recovering the exact share guarantee of Theorem 5.1. More generally, if the instantiation rule incurs $\eta_i \leq \varepsilon \cdot s(\widehat{V}_i, n)$ for all $i \in N$ (for some $\varepsilon \in [0, 1]$), then $\varepsilon_i \leq \varepsilon$ and we obtain the uniform guarantee $v_i(A_i) \geq (1 - \varepsilon) \, s(\widehat{V}_i, n)$ for every agent. When $s$ corresponds to an $\alpha$-approximation of a target benchmark (e.g., $s(V_i, n) = \alpha \cdot \text{MMS}_i$), this can be viewed as an additional multiplicative factor $(1 - \varepsilon_i)$ on top of $\alpha$.

## 6. Identical Valuations

Motivated by semi-online scheduling on *identical machines*, we also study the structured case where all agents share the same additive valuation function. Due to space constraints, we defer the full treatment of this setting to Appendix E. The main takeaway is that identical valuations are strong enough to recover an online EF1 guarantee even without any information about future goods (via the greedy rule that assigns each arriving good to a currently least-valued bundle), but they cannot circumvent the hardness of EFX: without future information no algorithm can achieve any positive approximation to EFX. With normalization information, we obtain a tight $\frac{\sqrt{5}-1}{2}$-EFX guarantee for two agents, while for $n \geq 3$ any positive approximation to EFX remains impossible.

## 7. Conclusion

We have explored the existence of standard notions of fairness in the online fair division setting with and without additional information. In the absence of future information, our results indicate that online algorithms cannot achieve fair outcomes. However, when the online algorithm is given normalization information or frequency predictions, (relatively) stronger fairness guarantees are possible. Under normalization information, we propose an algorithm is EF1 and 0.5-MMS for $n = 2$, and PROP1 for any $n$. Given frequency predictions, we introduce a meta-algorithm that leverages frequency predictions to match the best-known offline guarantees for a broad class of "share-based" fairness notions. Our complementary impossibility results across the multiple settings emphasizes the limitations of these additional information, despite its relative power.

The two-agent restrictions in our envy-based positive results should be read as structural boundaries of the corresponding information models, rather than as artifacts of the algorithms: our lower bounds rule out any positive approximation to EF1 for $n \geq 3$ under normalization information and to EFX for $n \geq 3$ even under frequency predictions. Similarly, our impossibility results already hold on additive instances and therefore extend to any richer valuation class containing additive valuations. The positive results, however, use additivity in an essential way: Algorithm 1 certifies fairness through bundle values plus a single outside good, while the frequency-prediction framework compresses future information into per-agent scalar multisets. Extending the positive results to submodular or subadditive valuations would therefore require new ideas.

Beyond the open questions raised in our work, several promising directions remain. It would be valuable to characterize what minimal forms of future information suffice for satisfying notions such as EF1 for all $n \geq 3$, and to extend the learning-augmented approach to other settings such as weaker adversaries or richer valuation classes.

## Acknowledgments

Nicholas Teh was supported by the Advanced Research + Invention Agency (ARIA) as part of the ASPAI project.

## Impact Statement

This paper presents work whose goal is to advance the theory of machine learning. There are many potential societal consequences of our work, none of which we feel must be specifically highlighted here.

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

# Appendix

## A. Limits of Randomization Under Adaptive Adversaries and Ex Post Guarantees

Throughout the paper we focus on deterministic online allocation algorithms and allow the adversary to be adaptive, i.e., to choose each future arrival after observing the algorithm's past allocation decisions. This is a standard and deliberately strong worst-case model in online fair division and semi-online scheduling.

It is natural to ask whether randomization can circumvent our impossibility results. In our setting, the fairness objective is evaluated *ex post* on the *realized* final allocation. Accordingly, when we discuss randomized online algorithms and say that such an algorithm *guarantees* a competitive ratio $\alpha$, we mean the following strong (pathwise) requirement: for every adaptive adversary strategy, *every* realization of the algorithm's internal randomness must yield a final allocation satisfying the $\alpha$-approximate fairness property.

Under this notion of ex post guarantees, deterministic lower bounds extend immediately to randomized algorithms, as we show with the following lemma.

**Lemma A.1.** *Consider any online fair division objective defined ex post on the final allocation (e.g., $\alpha$-EF1, $\alpha$-PROP1, or $\alpha$-MMS). If no deterministic online algorithm can guarantee a strictly positive competitive ratio against an adaptive adversary, then neither can any randomized online algorithm (with internal randomness) against the same adversary model.*

*Proof.* Assume for contradiction that there exist a constant $\alpha > 0$ and a randomized online algorithm $\mathcal{R}$ that guarantees competitive ratio at least $\alpha$ against an adaptive adversary, under the ex post (pathwise) guarantee described above. That is, for every adaptive adversary strategy $\mathcal{A}$ and every realization of $\mathcal{R}$'s internal randomness, the final allocation produced by the interaction satisfies the $\alpha$-approximate objective.

Model $\mathcal{R}$'s internal randomness by a (possibly infinite) random tape $\omega$. For each fixed tape $\omega$, define $\mathcal{R}^\omega$ to be the deterministic online algorithm obtained by running $\mathcal{R}$ with its random choices hardwired according to $\omega$.

Fix any adaptive adversary strategy $\mathcal{A}$. Consider running $\mathcal{A}$ against the randomized algorithm $\mathcal{R}$ with random tape $\omega$. By construction of $\mathcal{R}^\omega$, the sequence of allocation decisions made by $\mathcal{R}$ on tape $\omega$ is exactly the sequence of allocation decisions made by the deterministic algorithm $\mathcal{R}^\omega$ when interacting with $\mathcal{A}$: in both executions the adversary observes the same history, and the algorithm makes the same choice at each round. Thus, the realized final allocation is identical in the two executions.

By the assumed guarantee of $\mathcal{R}$, this realized final allocation satisfies the $\alpha$-approximate fairness objective. Since $\mathcal{A}$ was arbitrary, it follows that the deterministic algorithm $\mathcal{R}^\omega$ guarantees competitive ratio at least $\alpha$ against an adaptive adversary. This contradicts the premise that no deterministic online algorithm can guarantee any strictly positive competitive ratio against an adaptive adversary. Therefore no randomized online algorithm can guarantee a strictly positive competitive ratio in this model. $\square$

**Remark.** Randomization can become meaningful under weaker adversary models (e.g., an *oblivious* adversary that commits to the entire sequence in advance) and/or when fairness is evaluated *in expectation* (ex ante); we view such variants as an interesting direction for future work.

## B. Omitted Proofs from Section 3

**Proposition 3.1.** *For $n \geq 2$, without future information, there does not exist any deterministic online algorithm with a competitive ratio strictly larger than 0 with respect to* EF1.

*Proof.* Suppose for contradiction that there exist $\gamma \in (0, 1]$ and an online allocation algorithm $\mathcal{F}$ such that, on every online instance, the final allocation $\mathcal{A} = (A_1, \ldots, A_n)$ returned by $\mathcal{F}$ is $\gamma$-EF1. Fix a parameter $K > 1/\gamma$.

Let good $g_1$ have $v_i(g_1) = 1$ for all $i \in N$; and let $a \in N$ be the agent to whom $\mathcal{F}$ allocates $g_1$. Next, let good $g_2$ have

$$v_a(g_2) = K \quad \text{and} \quad v_j(g_2) = \frac{1}{K} \text{ for all } j \in N \setminus \{a\}.$$

If $\mathcal{F}$ allocates $g_2$ to agent $a$, then let one last good $g_3$ arrive with $v_i(g_3) = 0$ for all $i \in N$ and terminates. Pick any $b \in N \setminus \{a\}$. Then $v_b(A_b) = 0$, while $A_a$ contains both $g_1$ and $g_2$, so for every $g \in A_a$,

$$v_b(A_a \setminus \{g\}) \geq \min\{v_b(g_1), v_b(g_2)\} = \frac{1}{K} > 0.$$

Thus, there is no $g \in A_a$ such that $v_b(A_b) \geq \gamma \cdot v_b(A_a \setminus \{g\})$, contradicting that $\mathcal{A}$ is $\gamma$-EF1. Therefore $g_2$ must be allocated to some agent $b \in N \setminus \{a\}$.

Now let $n - 1$ further goods $g_3, \ldots, g_{n+1}$ arrive. For each $t \in \{3, \ldots, n+1\}$, define

$$v_a(g_t) = v_b(g_t) = K \quad \text{and} \quad v_j(g_t) = 1 \text{ for all } j \in N \setminus \{a, b\}.$$

We consider two cases.

**Case 1: Agent $b$ receives at least one good from $\{g_3, \ldots, g_{n+1}\}$.** If agent $a$ receives none of these goods, then $A_a = \{g_1\}$ and $A_b$ contains $g_2$ and at least one good from $\{g_3, \ldots, g_{n+1}\}$. Thus for every $g \in A_b$ we have $v_a(A_b \setminus \{g\}) \geq K$, so $\gamma$-EF1 for the pair $(a, b)$ implies $1 = v_a(A_a) \geq \gamma K$, i.e., $\gamma \leq 1/K$. This is a contradiction.

Thus, agent $a$ also receives at least one good from $\{g_3, \ldots, g_{n+1}\}$. If $n \geq 3$, then $a$ and $b$ together receive at least four goods ($g_1$, $g_2$, and one additional good each), leaving at most $n - 3$ goods for the $n - 2$ agents in $N \setminus \{a, b\}$. Therefore some agent $c \in N \setminus \{a, b\}$ receives no good. But $A_a$ contains $g_1$ and at least one good from $\{g_3, \ldots, g_{n+1}\}$, each valued 1 by agent $c$, so for every $g \in A_a$ we have $v_c(A_a \setminus \{g\}) \geq 1$ while $v_c(A_c) = 0$. Thus $\mathcal{A}$ is not $\gamma$-EF1, a contradiction. (If $n = 2$, this subcase cannot occur because $\{g_3, \ldots, g_{n+1}\}$ contains only one good.)

**Case 2: Agent $b$ receives no good from $\{g_3, \ldots, g_{n+1}\}$.** Then $A_b = \{g_2\}$, so $v_b(A_b) = 1/K$. The remaining $n$ goods are allocated among the $n - 1$ agents in $N \setminus \{b\}$, so some agent $i \in N \setminus \{b\}$ satisfies $|A_i| \geq 2$. Let $g^* \in \arg\max_{g \in A_i} v_b(g)$. Since valuations are additive, for any $g \in A_i$ we have $v_b(A_i \setminus \{g\}) = v_b(A_i) - v_b(g)$. Thus, $v_b(A_i \setminus \{g^*\}) = \min_{g \in A_i} v_b(A_i \setminus \{g\})$. Thus, if $\mathcal{A}$ is $\gamma$-EF1 for the ordered pair $(b, i)$, then removing $g^*$ from $A_i$ gives the weakest equirement among goods in $A_i$, so

$$v_b(A_b) \geq \gamma \cdot v_b(A_i \setminus \{g^*\}).$$

Now, $g_2 \notin A_i$ and every good $g \neq g_2$ satisfies $v_b(g) \geq 1$, so $A_i \setminus \{g^*\} \neq \varnothing$ and $v_b(A_i \setminus \{g^*\}) \geq 1$. Therefore,

$$\frac{1}{K} = v_b(A_b) \geq \gamma \cdot v_b(A_i \setminus \{g^*\}) \geq \gamma,$$

i.e., $\gamma \leq 1/K$, contradicting $K > 1/\gamma$.

In both cases we obtain a contradiction; thus, no online algorithm can guarantee any positive approximation factor with respect to EF1. $\square$

## C. Omitted Proofs from Section 4

**Theorem 4.1.** *For $n = 2$, given normalization information, Algorithm 1 returns an* EF1 *allocation.*

*Proof.* Let $A_i^t$ denote the bundle of agent $i$ after Algorithm 1 allocates the first $t$ goods, and let $A_i := A_i^m$ be the final bundle. For $n = 2$, write $j$ for the other agent. For each $t \in \{0, 1, \ldots, m\}$ define

$$M_i^t := \max_{g \in A_j^t} v_i(g),$$

with the convention that the maximum of an empty set is 0. Note that $M_i^t$ is nondecreasing in $t$.

In iteration $t$ (when good $g_t$ arrives), Algorithm 1 computes

$$x_i^t = v_i(A_i^{t-1}) + \tfrac{1}{2} \max\{M_i^{t-1}, v_i(g_t)\},$$

and removes agent $i$ from the active set whenever $x_i^t \geq \frac{1}{2}$.

We first show that for every agent $i$,

$$v_i(A_i) + \tfrac{1}{2}M_i^m \geq \tfrac{1}{2}. \tag{1}$$

After the removal step in iteration $t$, either the algorithm continues (so $g_t$ is allocated to the other agent $j$, hence $g_t \in A_j$ and therefore $M_i^m \geq \max\{M_i^{t-1}, v_i(g_t)\}$), or the early-stopping condition $N' = \varnothing$ triggers and all remaining goods (including $g_t$) are assigned to agent 1. In this early-stopping case, if $i \neq 1$ then $g_t \in A_1 \subseteq \bigcup_{\ell \neq i} A_\ell$, so again $M_i^m \geq \max\{M_i^{t-1}, v_i(g_t)\}$, whereas if $i = 1$ then $M_1^m \geq M_1^{t-1}$ and also $v_1(A_1) \geq v_1(A_1^{t-1}) + v_1(g_t)$. In all cases,

$$v_i(A_i) + \tfrac{1}{2}M_i^m \geq v_i(A_i^{t-1}) + \tfrac{1}{2}\max\{M_i^{t-1}, v_i(g_t)\} = x_i^t \geq \tfrac{1}{2},$$

which proves (1) for removed agents.

It remains to consider an agent $i$ that is never removed (i.e., remains active throughout). In this case Algorithm 1 never triggers the early stopping condition "$N' = \varnothing$", so all $m$ goods are allocated within the main loop. If $A_j = \varnothing$, then $M_i^m = 0$ and $v_i(A_i) = v_i(G) = 1$, so (1) holds trivially. Thus, assume $A_j \neq \varnothing$ and let $h$ be the last good allocated to agent $j$; denote by $t_h$ the iteration in which $h$ arrives. Because agent $j$ receives $h$ at iteration $t_h$, it must be active after the removal step in that iteration, and therefore $x_j(t_h) < \frac{1}{2}$. Dropping the maximum term in the definition of $x_j^{t_h}$ gives

$$v_j(A_j \setminus \{h\}) + \tfrac{1}{2}v_j(h) < \tfrac{1}{2}. \tag{2}$$

Since agent $i$ is active throughout, and every good in $A_j$ is allocated to $j$ while $i$ is active, Algorithm 1 must have had $v_j(g) \geq v_i(g)$ for all $g \in A_j$. Applying this to (2) gives us

$$v_i(A_j \setminus \{h\}) + \tfrac{1}{2}v_i(h) \leq v_j(A_j \setminus \{h\}) + \tfrac{1}{2}v_j(h) < \tfrac{1}{2}.$$

Using the fact that $v_i(G) = 1$ we have

$$1 = v_i(A_i) + v_i(A_j \setminus \{h\}) + v_i(h) < v_i(A_i) + (\tfrac{1}{2} - \tfrac{1}{2}v_i(h)) + v_i(h) = v_i(A_i) + \tfrac{1}{2} + \tfrac{1}{2}v_i(h).$$

Thus $v_i(A_i) + \tfrac{1}{2}v_i(h) > \tfrac{1}{2}$. Since $h \in A_j$, we have $v_i(h) \leq M_i^m$, implying (1).

If $A_j = \varnothing$, then agent $i$ trivially satisfies EF1 towards agent $j$. Otherwise, let $g^* \in \arg\max_{g \in A_j} v_i(g)$, so that $v_i(g^*) = M_i^m$. Inequality (1) is equivalent to $2v_i(A_i) + v_i(g^*) \geq 1$. Using $v_i(A_j) = v_i(G) - v_i(A_i) = 1 - v_i(A_i)$, this rearranges to

$$v_i(A_i) \geq v_i(A_j) - v_i(g^*) = v_i(A_j \setminus \{g^*\}),$$

which is exactly the EF1 requirement (from agent $i$ towards agent $j$). Since this holds for both agents, the allocation returned by Algorithm 1 is EF1. $\qquad\square$

**Theorem 4.2.** *For $n \geq 2$, given normalization information, Algorithm 1 returns a PROP1 allocation.*

*Proof.* Let $A_i^t$ denote the bundle of agent $i$ after Algorithm 1 allocates the first $t$ goods, and let $A_i := A_i^m$ be the final bundle. Fix $\alpha := \frac{n-1}{n}$. For each $t \in \{0, 1, \ldots, m\}$ define

$$M_i^t := \max_{g \in \cup_{j \neq i} A_j^t} v_i(g),$$

with the convention that the maximum of an empty set is 0. Note that $M_i^t$ is nondecreasing in $t$.

In iteration $t$ (when good $g_t$ arrives), Algorithm 1 computes

$$x_i^t = v_i(A_i^{t-1}) + \alpha \max\{M_i^{t-1}, v_i(g_t)\},$$

and removes agent $i$ from the active set whenever $x_i^t \geq \frac{1}{n}$.

We prove a stronger guarantee: for every agent $i$,

$$v_i(A_i) + \alpha M_i^m \geq \tfrac{1}{n}. \tag{3}$$

Since $\alpha \leq 1$, (3) implies PROP1: if $A_i = G$ we are done; otherwise take a good $g^* \in \arg\max_{g \in \cup_{j \neq i} A_j} v_i(g)$ so that $v_i(g^*) = M_i^m$, and note $v_i(A_i) + v_i(g^*) \geq v_i(A_i) + \alpha M_i^m \geq 1/n$.

**Case 1: agent $i$ becomes inactive during the algorithm.** Suppose agent $i$ is removed at iteration $t$, i.e.,

$$x_i^t = v_i(A_i^{t-1}) + \alpha \max\{M_i^{t-1}, v_i(g_t)\} \geq \frac{1}{n}.$$

After the removal step in iteration $t$, either the algorithm continues, in which case $g_t$ is allocated to some agent $k \neq i$ and therefore $g_t \in \bigcup_{j \neq i} A_j$, implying $M_i^m \geq \max\{M_i^{t-1}, v_i(g_t)\}$; or the early-stopping condition $N' = \varnothing$ triggers and all remaining goods (including $g_t$) are assigned to agent 1. In this case, if $i \neq 1$ then $g_t \in A_1 \subseteq \bigcup_{j \neq i} A_j$, so $M_i^m \geq \max\{M_i^{t-1}, v_i(g_t)\}$, whereas if $i = 1$ then $M_1^m \geq M_1^{t-1}$ and also $v_1(A_1) \geq v_1(A_1^{t-1}) + v_1(g_t)$. In all cases,

$$v_i(A_i) + \alpha M_i^m \geq v_i(A_i^{t-1}) + \alpha \max\{M_i^{t-1}, v_i(g_t)\} = x_i^t \geq \frac{1}{n},$$

proving (3) for inactive agents.

**Case 2: agent $i$ remains active throughout.** Now suppose agent $i$ is never removed, i.e., it is active in every iteration. In particular, Algorithm 1 never triggers the early stopping condition "$N' = \varnothing$", so all $m$ goods are allocated within the main loop.

For each agent $j \neq i$, define $h_j$ as follows. If $A_j = \varnothing$, let $h_j$ be a dummy good satisfying $v_\ell(h_j) = 0$ for every agent $\ell$; then (4) holds trivially. Otherwise, let $h_j$ denote the last good allocated to agent $j$, and denote by $t_j$ the iteration in which $h_j$ arrives. Because agent $j$ receives $h_j$ at iteration $t_j$, it must be active after the removal step in that iteration, and therefore $x_j^{t_j} < \frac{1}{n}$. Dropping the maximum term in the definition of $x_j^{t_j}$ gives

$$v_j(A_j \setminus \{h_j\}) + \alpha v_j(h_j) < \tfrac{1}{n}. \tag{4}$$

Since agent $i$ is active throughout, every good in $A_j$ is allocated to $j$ while $i$ is active. Therefore, for all $g \in A_j$ we must have $v_j(g) \geq v_i(g)$ (otherwise Algorithm 1 would have allocated $g$ to $i$ instead of $j$). Applying this to (4) yields

$$v_i(A_j \setminus \{h_j\}) + \alpha v_i(h_j) < \tfrac{1}{n} \qquad \text{for every } j \neq i. \tag{5}$$

Summing (5) over all $j \neq i$ gives

$$\sum_{j \neq i} v_i(A_j \setminus \{h_j\}) + \alpha \sum_{j \neq i} v_i(h_j) < \tfrac{n-1}{n}. \tag{6}$$

On the other hand, since $v_i(G) = 1$,

$$1 = v_i(A_i) + \sum_{j \neq i} v_i(A_j \setminus \{h_j\}) + \sum_{j \neq i} v_i(h_j). \tag{7}$$

Combining (6) and (7) gives us

$$1 < v_i(A_i) + \tfrac{n-1}{n} + (1-\alpha) \sum_{j \neq i} v_i(h_j) = v_i(A_i) + \tfrac{n-1}{n} + \tfrac{1}{n} \sum_{j \neq i} v_i(h_j),$$

and hence

$$v_i(A_i) + \tfrac{1}{n} \sum_{j \neq i} v_i(h_j) > \tfrac{1}{n}. \tag{8}$$

Since there are $n - 1$ terms in the sum, we have $\sum_{j \neq i} v_i(h_j) \leq (n-1) \cdot \max_{j \neq i} v_i(h_j)$, so (8) implies

$$v_i(A_i) + \alpha \cdot \max_{j \neq i} v_i(h_j) > \tfrac{1}{n}.$$

Finally, for every $j \neq i$ with $A_j \neq \varnothing$ we have $h_j \in \cup_{k \neq i} A_k$, while for $j$ with $A_j = \varnothing$ we have $v_i(h_j) = 0$ by construction. Thus, $\max_{j \neq i} v_i(h_j) \leq M_i^m$. Therefore $v_i(A_i) + \alpha M_i^m \geq v_i(A_i) + \alpha \cdot \max_{j \neq i} v_i(h_j) > 1/n$, which proves (3).

Since (3) holds for every agent $i$, Algorithm 1 returns a PROP1 allocation. $\qquad \square$

**Theorem 4.3.** *For $n \geq 2$, given normalization information, there does not exist any online algorithm with a competitive ratio strictly larger than $0$ with respect to* EFX.

*Proof.* Suppose for a contradiction that there exists a $\gamma$-competitive algorithm for approximating EFX given normalization information, for some $\gamma \in (0, 1]$. Let $k \in \mathbb{N}$ be large. Consider the case of $n = 2$. Let the first two goods be $g_1$ and $g_2$, with $v_1(g_1) = v_2(g_1) = 2^{-k^3}, v_1(g_2) = 2^{-k^2}$, and $v_2(g_2) = 2^{-k^3}$. Assume w.l.o.g. that $g_1$ is allocated to agent 2. If $g_2$ is allocated to agent 1, let the next (and final) good be $g_3$ with valuations as follows:

| v | $g_1$ | $g_2$ | $g_3$ |
|---|---|---|---|
| 1 | $2^{-k^3}$ | $\left(2^{-k^2}\right)$ | $1 - 2^{-k^3} - 2^{-k^2}$ |
| 2 | $\left(2^{-k^3}\right)$ | $2^{-k^3}$ | $1 - 2^{-k^3+1}$ |

Then, if $g_3$ is allocated to agent 1, we have that $v_2(A_2) = 2^{-k^3}$ and $v_2(A_1 \setminus \{g_2\}) = 1 - 2^{-k^3+1}$ (observe that $v_2(g_2) < v_2(g_3)$ for sufficiently large $k$). This gives us

$$\gamma \leq \frac{2^{-k^3}}{1 - 2^{-k^3+1}} = \frac{1}{2^{k^3} - 2} \to 0$$

as $k \to \infty$, giving us a contradiction. On the other hand, if $g_3$ is allocated to agent 2, then $v_1(A_1) = 2^{-k^2}$ and $v_1(A_2 \setminus \{g_1\}) = v_1(g_3) = 1 - 2^{-k^3} - 2^{-k^2}$ (observe that $v_1(g_1) < v_1(g_3)$ for some sufficiently large $k$). This gives us

$$\gamma \leq \frac{2^{-k^2}}{1 - 2^{-k^3} - 2^{-k^2}} = \frac{1}{2^{k^2} - 2^{-k^3+k^2} - 1} \to 0$$

as $k \to \infty$, giving us a contradiction. Thus, $g_2$ cannot be allocated to agent 1. Let it be allocated to agent 2. Then, let each successive good $g$ that arrives have value $v_1(g) = 2^{-k^2}$ and $v_2(g) = 2^{-k^3}$ (up to a maximum of $\ell - 2 = 2^{k^2} - 2$ such goods). Consider the earliest point in time where the algorithm allocates a good that arrives to agent 1. If it exists, let such a good be $g_j$ (i.e., the $j$-th good that arrives overall). We split our analysis into two cases.

**Case 1:** $\{g_3, \dots, g_\ell\} \subset A_2$. This means that none of the goods that arrived as described above was allocated to agent 1. Let the $(\ell + 1)$-th and final good be $g_{\ell+1}$, with valuations as follows:

| v | $g_1$ | $g_2$ | $\cdots$ | $g_\ell$ | $g_{\ell+1}$ |
|---|---|---|---|---|---|
| 1 | $2^{-k^3}$ | $2^{-k^2}$ | $\cdots$ | $2^{-k^2}$ | $2^{-k^2} - 2^{-k^3}$ |
| 2 | $\left(2^{-k^3}\right)$ | $\left(2^{-k^3}\right)$ | $\cdots$ | $\left(2^{-k^3}\right)$ | $1 - 2^{-k^3+k^2}$ |

If $g_{\ell+1}$ is also allocated to agent 2, then $v_1(A_1) = 0$ and the allocation violates $\gamma$-EFX for every $\gamma > 0$, and thus the competitive ratio is 0, a contradiction. Thus, $g_{\ell+1}$ must be allocated to agent 1. We have that $v_1(A_1) = 2^{-k^2} - 2^{-k^3}$ and $v_1(A_2) \setminus \{g_1\}) = (2^{k^2} - 1) \cdot 2^{-k^2} = 1 - 2^{-k^2}$ (observe that $v_1(g_1) = \min_{g \in A_2} v_1(g)$ for some sufficiently large $k$). This gives us

$$\gamma \leq \frac{2^{-k^2} - 2^{-k^3}}{1 - 2^{-k^2}} = \frac{1 - 2^{-k^3+k^2}}{2^{k^2} - 1} \to 0$$

as $k \to \infty$, giving us a contradiction.

**Case 2:** $\exists j : j \leq \ell$ such that $g_j \in A_1$. This means that $\{g_3, \dots, g_{j-1}\} \subset A_2$. Let the $(j + 1)$-th (and final) good be $g_{j+1}$, with valuations as follows:

| v | $g_1$ | $g_2$ | $\cdots$ | $g_{j-1}$ | $g_j$ | $g_{j+1}$ |
|---|---|---|---|---|---|---|
| 1 | $2^{-k^3}$ | $2^{-k^2}$ | $\cdots$ | $2^{-k^2}$ | $\left(2^{-k^2}\right)$ | $1 - 2^{-k^3} - (j - 1) \cdot 2^{-k^2}$ |
| 2 | $\left(2^{-k^3}\right)$ | $\left(2^{-k^3}\right)$ | $\cdots$ | $\left(2^{-k^3}\right)$ | $2^{-k^3}$ | $1 - j2^{-k^3}$ |

Note that $k^3 - 1 > k^2$ for some sufficiently large $k > 1$. Equivalently, we get (via algebraic manipulation) that

$$2^{-k^3+1} < 2^{-k^2}. \tag{9}$$

Then, since $j \le \ell = 2^{k^2}$, algebraic manipulation gives us $1 - (j-1) \cdot 2^{-k^2} \ge 2^{-k^2}$. Combining this with (9) gives us

$$1 - 2^{-k^3} - (j-1) \cdot 2^{-k^2} > 2^{-k^3}. \tag{10}$$

Now, if $g_{j+1}$ is allocated to agent 2, then $v_1(A_1) = 2^{-k^2}$ and $v_1(A_2 \setminus \{g_1\}) = 1 - 2^{-k^3} - 2^{-k^2}$ (note that by (10), we have that $v_1(g_1) < v_1(g_{j+1})$). This gives us

$$\gamma \le \frac{2^{-k^2}}{1 - 2^{-k^3} - 2^{-k^2}} = \frac{1}{2^{k^2} - 2^{-k^3+k^2} - 1} \to 0$$

as $k \to \infty$, giving us a contradiction.

If instead $g_{j+1}$ is allocated to agent 1, then $A_1 = \{g_j, g_{j+1}\}$ and $A_2 = \{g_1, g_2, \ldots, g_{j-1}\}$. Hence

$$v_2(A_2) = (j-1) \cdot 2^{-k^3} \quad \text{and} \quad v_2(A_1 \setminus \{g_j\}) = v_2(g_{j+1}) = 1 - j \cdot 2^{-k^3}.$$

By $\gamma$-EFX applied to the ordered pair $(2, 1)$ and the good $g_j \in A_1$,

$$\gamma \le \frac{v_2(A_2)}{v_2(A_1 \setminus \{g_j\})} = \frac{(j-1) \cdot 2^{-k^3}}{1 - j \cdot 2^{-k^3}} \le \frac{2^{-k^3+k^2}}{1 - 2^{-k^3+k^2}} \to 0 \qquad \text{as } k \to \infty,$$

where we used $j \le \ell = 2^{k^2}$. This is again a contradiction.

In all cases, we arrive at a contradiction, thereby proving our result.

For the case of $n \ge 3$, the claim follows from Theorem E.4, which proves 0-competitiveness for EFX even under identical valuations (a special case of the present model). □

**Theorem 4.4.** *For $n \ge 3$, given normalization information, there does not exist any online algorithm with a competitive ratio strictly larger than 0 with respect to* EF1.

*Proof.* Suppose for a contradiction that there exists a $\gamma$-competitive algorithm for approximating EF1 given normalization information, for some $\gamma \in (0, 1]$. Let $\varepsilon > 0$ be a sufficiently small constant and $\delta > 1$ be a large constant. Let the first two goods be $g_1, g_2$ with valuations as follows:

| **v** | $g_1$ | $g_2$ |
|---|---|---|
| 1 | $\varepsilon$ | $\frac{1}{n}$ |
| 2 | $\varepsilon$ | $\frac{\varepsilon}{\delta}$ |
| 3 | $\varepsilon$ | $\frac{\varepsilon}{\delta}$ |
| $\vdots$ | $\vdots$ | $\vdots$ |
| $n$ | $\varepsilon$ | $\frac{\varepsilon}{\delta}$ |

Without loss of generality, we assume that $g_1$ is allocated to agent 1. If $g_2$ is allocated to agent 1, let the next $n - 1$ (and final) goods be $g_3, \ldots, g_{n+1}$ with the following valuations:

| **v** | $g_1$ | $g_2$ | $g_3$ | $\cdots$ | $g_n$ | $g_{n+1}$ |
|---|---|---|---|---|---|---|
| 1 | $\textcircled{\varepsilon}$ | $\textcircled{\frac{1}{n}}$ | $\frac{1}{n} + \varepsilon$ | $\cdots$ | $\frac{1}{n} + \varepsilon$ | $\frac{1}{n} - (n-1)\varepsilon$ |
| 2 | $\varepsilon$ | $\frac{\varepsilon}{\delta}$ | $\frac{\varepsilon}{\delta^2}$ | $\cdots$ | $\frac{\varepsilon}{\delta^2}$ | $1 - \varepsilon - \frac{\varepsilon}{\delta} - (n-2)\frac{\varepsilon}{\delta^2}$ |
| 3 | $\varepsilon$ | $\frac{\varepsilon}{\delta}$ | $\frac{\varepsilon}{\delta^2}$ | $\cdots$ | $\frac{\varepsilon}{\delta^2}$ | $1 - \varepsilon - \frac{\varepsilon}{\delta} - (n-2)\frac{\varepsilon}{\delta^2}$ |
| $\vdots$ | $\vdots$ | $\vdots$ | $\vdots$ | | $\vdots$ | $\vdots$ |
| $n$ | $\varepsilon$ | $\frac{\varepsilon}{\delta}$ | $\frac{\varepsilon}{\delta^2}$ | $\cdots$ | $\frac{\varepsilon}{\delta^2}$ | $1 - \varepsilon - \frac{\varepsilon}{\delta} - (n-2)\frac{\varepsilon}{\delta^2}$ |

Suppose each good in $\{g_3, \ldots, g_{n+1}\}$ is allocated to a unique agent in $N \setminus \{1\}$ (i.e., each agent in $N \setminus \{1\}$ receives exactly one good). Then, consider the agent that received $g_3$, let it be agent $i$. However, $v_i(A_i) = v_i(g_3) = \frac{\varepsilon}{\delta^2}$ and $v_i(A_1 \setminus \{g_1\}) = \frac{\varepsilon}{\delta}$, giving us

$$\gamma \leq \frac{\frac{\varepsilon}{\delta^2}}{\frac{\varepsilon}{\delta}} = \frac{1}{\delta} \to 0$$

as $\delta \to \infty$, a contradiction. Thus, we must have that either some good in $\{g_3, \ldots, g_{n+1}\}$ is allocated to agent 1 and/or some agent in $N \setminus \{1\}$ receives strictly more than one good in the set. However, in either of these cases, by the pigeonhole principle, there must exist an agent in $N \setminus \{1\}$ that receives no good, leading to an allocation whose EF1-approximation factor is 0. Hence, $g_2$ cannot be allocated to agent 1.

W.l.o.g., let $g_2$ be allocated to agent 2, and the next $n - 1$ (and final) goods be $g_3, \ldots, g_{n+1}$ with the following valuations:

| $\mathbf{v}$ | $g_1$ | $g_2$ | $g_3$ | $\cdots$ | $g_n$ | $g_{n+1}$ |
|---|---|---|---|---|---|---|
| 1 | $\textcircled{\varepsilon}$ | $\frac{1}{n}$ | $\frac{1}{n} + \varepsilon$ | $\cdots$ | $\frac{1}{n} + \varepsilon$ | $\frac{1}{n} - (n-1)\varepsilon$ |
| 2 | $\varepsilon$ | $\textcircled{\frac{\varepsilon}{\delta}}$ | $\varepsilon$ | $\cdots$ | $\varepsilon$ | $1 - (n-1)\varepsilon - \frac{\varepsilon}{\delta}$ |
| 3 | $\varepsilon$ | $\frac{\varepsilon}{\delta}$ | $\varepsilon$ | $\cdots$ | $\varepsilon$ | $1 - (n-1)\varepsilon - \frac{\varepsilon}{\delta}$ |
| $\vdots$ | $\vdots$ | $\vdots$ | $\vdots$ | $\cdots\cdots$ | $\vdots$ | $\vdots$ |
| $n$ | $\varepsilon$ | $\frac{\varepsilon}{\delta}$ | $\varepsilon$ | $\cdots$ | $\varepsilon$ | $1 - (n-1)\varepsilon - \frac{\varepsilon}{\delta}$ |

If agent 1 and 2 collectively is allocated at least two goods from $\{g_3, \ldots, g_{n+1}\}$, then by the pigeonhole principle, there exists some agent in $N \setminus \{1, 2\}$ that will receive no good, and we get a 0-EF1 allocation, a contradiction. Thus, we assume that agent 1 and 2 collectively is allocated at most one good from $\{g_3, \ldots, g_{n+1}\}$. We split our analysis into two cases.

**Case 1:** $|(A_1 \cup A_2) \cap \{g_3, \ldots, g_{n+1}\}| = 1$. Let such a good in the intersection be $g^*$. If $g^*$ was allocated to agent 1, then $v_2(A_2) = \varepsilon/\delta$ and $v_2(A_1 \setminus \{g^*\}) = v_2(g_1) = \varepsilon$ (for sufficiently small $\varepsilon$). Consequently we get $\gamma \leq (\varepsilon/\delta)/\varepsilon = 1/\delta \to 0$ as $\delta \to \infty$, a contradiction.

Thus, we must have that $g^*$ is allocated to agent 2. Then, $v_1(A_1) = \varepsilon$ and for every $g \in A_2$ we have $v_1(A_2 \setminus \{g\}) \geq \frac{1}{n} - (n-1)\varepsilon$, giving us

$$\gamma \leq \frac{\varepsilon}{\frac{1}{n} - (n-1)\varepsilon} \to 0$$

as $\varepsilon \to 0$, a contradiction.

**Case 2:** $|(A_1 \cup A_2) \cap \{g_3, \ldots, g_{n+1}\}| = 0$. Then, by the pigeonhole principle, there exists some agent $i \in N \setminus \{1, 2\}$ that receives at least two goods from $\{g_3, \ldots, g_{n+1}\}$. Then, $v_2(A_2) = v_2(g_2) = \frac{\varepsilon}{\delta}$ and, since $|A_i| \geq 2$ and every good in $A_i$ has $v_2$-value at least $\varepsilon$ (for sufficiently small $\varepsilon$), we have $v_2(A_i \setminus \{g\}) \geq \varepsilon$ for every $g \in A_i$. Consequently, we get that

$$\gamma \leq \frac{\frac{\varepsilon}{\delta}}{\varepsilon} = \frac{1}{\delta} \to 0$$

as $\delta \to \infty$, a contradiction.

In both cases, we arrive at a contradiction, and our result follows. $\qquad\square$

**Theorem 4.5.** *Let $\mathcal{A}$ be the allocation returned by Algorithm 1 when run on the scaled valuations $\widetilde{v}_i := v_i/\underline{T}_i$.*

(i) *If $n = 2$, for every $i, j \in N$ with $A_j \neq \varnothing$, there exists a good $g \in A_j$ such that $v_i(A_i) \geq v_i(A_j \setminus \{g\}) - \rho_i$.*

(ii) *For any $n \geq 2$ and every $i \in N$, either $A_i = G$ or there exists a good $g \in G \setminus A_i$ such that $v_i(A_i \cup \{g\}) \geq \kappa_i \cdot \frac{v_i(G)}{n}$. In particular, letting $\kappa := \min_{i \in N} \kappa_i$, the allocation is $\kappa$-PROP1.*

*Proof.* Fix an instance and intervals $([\underline{T}_i, \overline{T}_i])_{i \in N}$. Run Algorithm 1 on $(\widetilde{v}_i)_{i \in N}$ and let $\mathcal{A}$ be the output allocation. For each $t \in \{0, 1, \ldots, m\}$, let $A_i^t$ denote agent $i$'s bundle after the first $t$ goods have been allocated (so $A_i^0 = \varnothing$ and $A_i^m = A_i$). Define the outside-max under $\widetilde{v}_i$ after $t$ allocations as

$$\widetilde{M}_i^t := \max\left\{\widetilde{v}_i(g) : g \in \bigcup_{j \neq i} A_j^t\right\}, \text{ and } \max \varnothing := 0.$$

Let $\alpha := \frac{n-1}{n}$.

We first prove that for every agent $i \in N$,

$$\widetilde{v}_i(A_i) + \alpha \cdot \widetilde{M}_i^m \geq \frac{1}{n}. \tag{11}$$

We split our analysis into two cases.

**Case 1: agent $i$ is removed from the active set at some iteration.** Let $t$ be the first iteration in which Algorithm 1 removes $i$, i.e.,

$$\widetilde{x}_i^t := \widetilde{v}_i(A_i^{t-1}) + \alpha \cdot \max\{\widetilde{M}_i^{t-1}, \widetilde{v}_i(g_t)\} \geq \frac{1}{n}.$$

After the removal step in iteration $t$, either the algorithm continues, in which case $g_t$ is allocated to some agent $\neq i$ and thus $\widetilde{M}_i^m \geq \max\{\widetilde{M}_i^{t-1}, \widetilde{v}_i(g_t)\}$; or the early-stopping condition triggers and all remaining goods $\{g_t, \ldots, g_m\}$ are assigned to agent 1. In the early stopping branch, if $i \neq 1$ then again $g_t \notin A_i$ and thus $\widetilde{M}_i^m \geq \max\{\widetilde{M}_i^{t-1}, \widetilde{v}_i(g_t)\}$. If instead $i = 1$, then $g_t \in A_1$ and we have that

$$\begin{aligned}
\widetilde{v}_1(A_1) + \alpha\widetilde{M}_1^m &\geq \widetilde{v}_1(A_1^{t-1}) + \widetilde{v}_1(g_t) + \alpha\widetilde{M}_1^{t-1} \\
&\geq \widetilde{v}_1(A_1^{t-1}) + \alpha \max\{\widetilde{M}_1^{t-1}, \widetilde{v}_1(g_t)\} \\
&= \widetilde{x}_1^t.
\end{aligned}$$

In all sub-cases,

$$\begin{aligned}
\widetilde{v}_i(A_i) + \alpha\widetilde{M}_i^m &\geq \widetilde{v}_i(A_i^{t-1}) + \alpha \max\{\widetilde{M}_i^{t-1}, \widetilde{v}_i(g_t)\} \\
&= \widetilde{x}_i^t \geq \frac{1}{n},
\end{aligned}$$

establishing (11).

**Case 2: agent $i$ is never removed (i.e., stays active throughout).** Then Algorithm 1 never triggers early stopping, so all $m$ goods are allocated inside the main loop. For each $j \neq i$ with $A_j \neq \varnothing$, let $h_j$ be the last good allocated to agent $j$ and let $t_j$ be its arrival time. Since $j$ receives $h_j$ at iteration $t_j$, it must still be active after the removal step in that iteration, and thus

$$\widetilde{x}_j^{t_j} = \widetilde{v}_j(A_j \setminus \{h_j\}) + \alpha \cdot \max\{\widetilde{M}_j^{t_j-1}, \widetilde{v}_j(h_j)\} < \frac{1}{n}.$$

Thus, we have that

$$\widetilde{v}_j(A_j \setminus \{h_j\}) + \alpha\widetilde{v}_j(h_j) < \frac{1}{n}. \tag{12}$$

For agents $j \neq i$ with $A_j = \varnothing$, interpret $h_j$ as a dummy good with $\widetilde{v}_\ell(h_j) = 0$ for all $\ell$, so (12) holds trivially.

Because $i$ is active throughout, every good in $A_j$ is allocated to $j$ at a time when $i \in N'$; since Algorithm 1 allocates each arriving good to an agent in $\arg\max_{k \in N'} \widetilde{v}_k(g)$, it follows that for every $g \in A_j$ we have $\widetilde{v}_j(g) \geq \widetilde{v}_i(g)$. By additivity, replacing $\widetilde{v}_j$ by $\widetilde{v}_i$ can only decrease the left-hand side of (12), giving us

$$\widetilde{v}_i(A_j \setminus \{h_j\}) + \alpha\widetilde{v}_i(h_j) < \frac{1}{n}. \tag{13}$$

for every $j \neq i$. Summing (13) over all $j \neq i$ gives us

$$\sum_{j \neq i} \widetilde{v}_i(A_j \setminus \{h_j\}) + \alpha \sum_{j \neq i} \widetilde{v}_i(h_j) < \frac{n-1}{n} = \alpha. \tag{14}$$

On the other hand, by additivity and completeness,

$$\widetilde{v}_i(G) = \widetilde{v}_i(A_i) + \sum_{j \neq i} \widetilde{v}_i(A_j \setminus \{h_j\}) + \sum_{j \neq i} \widetilde{v}_i(h_j). \tag{15}$$

Combining (14) and (15) gives us

$$\widetilde{v}_i(G) < \widetilde{v}_i(A_i) + \alpha + (1 - \alpha) \sum_{j \neq i} \widetilde{v}_i(h_j)$$

$$= \widetilde{v}_i(A_i) + \alpha + \frac{1}{n} \sum_{j \neq i} \widetilde{v}_i(h_j),$$

and therefore

$$\widetilde{v}_i(A_i) + \frac{1}{n} \sum_{j \neq i} \widetilde{v}_i(h_j) > \widetilde{v}_i(G) - \alpha. \tag{16}$$

Since $\widetilde{v}_i(G) = T_i/\underline{T}_i \geq 1$ and $\alpha = \frac{n-1}{n}$, we have $\widetilde{v}_i(G) - \alpha \geq 1 - \alpha = \frac{1}{n}$, so (16) implies

$$\widetilde{v}_i(A_i) + \frac{1}{n} \sum_{j \neq i} \widetilde{v}_i(h_j) > \frac{1}{n}.$$

Finally, $\sum_{j \neq i} \widetilde{v}_i(h_j) \leq (n-1) \max_{j \neq i} \widetilde{v}_i(h_j) \leq (n-1) \widetilde{M}_i^m$, so the left-hand side is at most $\widetilde{v}_i(A_i) + \alpha \widetilde{M}_i^m$. This proves (11).

Next, we prove (ii). Fix any agent $i \in N$. If $A_i = G$, we are done. Otherwise, let $g^* \in G \setminus A_i$ maximize $\widetilde{v}_i(\cdot)$ among outside goods, so $\widetilde{v}_i(g^*) = \widetilde{M}_i^m$. Using (11) and $\alpha \leq 1$,

$$\widetilde{v}_i(A_i \cup \{g^*\}) = \widetilde{v}_i(A_i) + \widetilde{v}_i(g^*)$$

$$= \widetilde{v}_i(A_i) + \widetilde{M}_i^m \geq \widetilde{v}_i(A_i) + \alpha \widetilde{M}_i^m \geq \frac{1}{n}.$$

Multiplying by $\underline{T}_i$ gives

$$v_i(A_i \cup \{g^*\}) \geq \frac{\underline{T}_i}{n}.$$

Since $T_i = v_i(G) \leq \overline{T}_i$, we have

$$\frac{\underline{T}_i}{n} \geq \frac{\underline{T}_i}{\overline{T}_i} \cdot \frac{T_i}{n} = \kappa_i \cdot \frac{v_i(G)}{n}.$$

Finally, we prove (i). Assume $n = 2$. Fix $i \in \{1, 2\}$ and let $j \neq i$. If $A_j = \varnothing$, then the EF1 requirement from $i$ to $j$ is vacuous. Otherwise, let $g^* \in A_j$ maximize $\widetilde{v}_i(\cdot)$ over $A_j$; then $\widetilde{v}_i(g^*) = \widetilde{M}_i^m$. For $n = 2$ we have $\alpha = \frac{1}{2}$, and (11) gives $\widetilde{v}_i(A_i) + \frac{1}{2} \widetilde{M}_i^m \geq \frac{1}{2}$, i.e.,

$$2\widetilde{v}_i(A_i) + \widetilde{v}_i(g^*) \geq 1.$$

Using additivity and $\widetilde{v}_i(G) = T_i/\underline{T}_i$, we have $\widetilde{v}_i(A_j \setminus \{g^*\}) = \widetilde{v}_i(G) - \widetilde{v}_i(A_i) - \widetilde{v}_i(g^*)$, so

$$\widetilde{v}_i(A_i) - \widetilde{v}_i(A_j \setminus \{g^*\}) = 2\widetilde{v}_i(A_i) + \widetilde{v}_i(g^*) - \widetilde{v}_i(G)$$

$$\geq 1 - \widetilde{v}_i(G).$$

Rearranging gives us

$$\widetilde{v}_i(A_i) \geq \widetilde{v}_i(A_j \setminus \{g^*\}) - (\widetilde{v}_i(G) - 1).$$

Multiplying by $\underline{T}_i$ gives

$$v_i(A_i) \geq v_i(A_j \setminus \{g^*\}) - (\widetilde{v}_i(G) - 1)\underline{T}_i = v_i(A_j \setminus \{g^*\}) - (T_i - \underline{T}_i).$$

Since $T_i \leq \overline{T}_i = \underline{T}_i + \rho_i$, we have $T_i - \underline{T}_i \leq \rho_i$, proving the result. $\qquad \square$

## C.1. Remark on the Additive EF1 Guarantee for $n = 2$ in Theorem 4.5

The additive EF1 guarantee in Theorem 4.5(i) cannot, in general, be restated as an $\alpha$-EF1 guarantee for any $\alpha > 0$ (even for $n = 2$), because under noisy advice an agent may end up with *zero* value while still satisfying the additive inequality. In contrast, PROP1 admits the clean multiplicative guarantee in Theorem 4.5(ii). We illustrate this gap in Example C.1 below.

*Example* C.1. Fix any $\varepsilon \in (0, 1)$ and consider $n = 2$ agents and two goods $g_1, g_2$ arriving in this order. Let the true valuations be

$$v_1(g_1) = 1, v_1(g_2) = 0 \qquad \text{and} \qquad v_2(g_1) = 1 - \varepsilon, v_2(g_2) = \varepsilon,$$

so $T_1 = T_2 = 1$. Give the algorithm the certified intervals

$$[\underline{T}_1, \overline{T}_1] = [1, 1] \qquad \text{and} \qquad [\underline{T}_2, \overline{T}_2] = [1 - \varepsilon, 1],$$

so $\rho_2 = \varepsilon$. Run Algorithm 1 on the scaled valuations $\widetilde{v}_i = v_i/\underline{T}_i$. At time $t = 1$, we have $\widetilde{v}_1(g_1) = 1$ and $\widetilde{v}_2(g_1) = (1 - \varepsilon)/(1 - \varepsilon) = 1$. Since $n = 2$, Algorithm 1 computes for both agents

$$\widetilde{x}_i^1 = \widetilde{v}_i(\varnothing) + \frac{1}{2}\max\{0, \widetilde{v}_i(g_1)\} = \frac{1}{2},$$

and thus removes both agents from the active set, triggering the early-stopping rule and assigning *all remaining goods* $\{g_1, g_2\}$ to agent 1. Hence $A_1 = \{g_1, g_2\}$ and $A_2 = \varnothing$.

This output is *not* $\alpha$-EF1 for any $\alpha > 0$: indeed $v_2(A_2) = 0$, while for either $g \in A_1$ we have $v_2(A_1 \setminus \{g\}) > 0$, so $0 \geq \alpha \cdot v_2(A_1 \setminus \{g\})$ fails. Nevertheless, the additive EF1 guarantee of Theorem 4.5(i) holds for agent 2: choosing $g = g_1$ gives us

$$v_2(A_2) = 0 \geq v_2(A_1 \setminus \{g_1\}) - \rho_2 = v_2(\{g_2\}) - \varepsilon = \varepsilon - \varepsilon = 0.$$

Thus, even for $n = 2$, robustness for EF1 in this advice model is naturally *additive*, whereas PROP1 admits the multiplicative factor $\kappa_i = \underline{T}_i/\overline{T}_i$.

# D. Omitted Proofs from Section 5

**Theorem 5.1.** *Let $s$ be a feasible share in the single-shot fair division setting. Then, in the online fair division setting, there exists an online algorithm using frequency predictions that guarantees each agent a bundle value of at least $s(V_i, n)$.*

*Proof.* Let $g_1, \ldots, g_m$ be the order of goods that arrive in the online setting. Fix some $t \in [m]$. Given an agent $i \in N$, let $V_i^t$ denote the frequency multiset for goods in $\{g_t, \ldots, g_m\}$. We note that $V_i^1 = V_i$ is agents $i$'s frequency multiset for all goods that will arrive. We say that a valuation function $v_i$ of agent $i$ is *consistent* with $V_i^t$ if $V_i^t = \{v_i(g_t), \ldots, v_i(g_m)\}$. Let $v_i^{t,\mathsf{IDO}}$ be a valuation function of agent $i$ that is consistent with $V_i^t$ and where for each $t \leq j \leq j'$, $v_i^{t,\mathsf{IDO}}(g_j) \geq v_i^{t,\mathsf{IDO}}(g_{j'})$, with the valuation profile $\mathbf{v}^{t,\mathsf{IDO}} = (v_1^{t,\mathsf{IDO}}, \ldots, v_n^{t,\mathsf{IDO}})$.[9] We fix $v_i^{t,\mathsf{IDO}}$ deterministically as follows: sort the multiset $V_i^t$ in non-increasing order (breaking ties arbitrarily but consistently), and assign the $k$-th largest value to good $g_{t+k-1}$ for each $k \in [m - t + 1]$.

Now, consider a single-shot fair division instance with goods $\{g_t, \ldots, g_m\}$. Let $\pi^t = (a_t, \ldots, a_m)$ be a *picking sequence* over goods $\{g_t, \ldots, g_m\}$, whereby $a_i \in N$ for each $i \in \{t, \ldots, m\}$, and agents take turns (according to $\pi^t$) picking their highest-valued good (according to their valuation function) from among the remaining goods available, breaking ties in a consistent manner (say, in favor of lower-indexed goods). Then, given a valuation profile $\mathbf{v} = (v_1, \ldots, v_n)$ and picking sequence $\pi^t$, we denote $\mathcal{A}^{\pi^t, \mathbf{v}} = \left(A_1^{\pi^t, \mathbf{v}}, \ldots, A_n^{\pi^t, \mathbf{v}}\right)$ as the allocation returned in the single-shot instance over the set of goods $\{g_t, \ldots, g_m\}$ induced by the picking sequence $\pi^t$ and valuation profile $\mathbf{v}$.

We then obtain the following result, which will be useful later on.

**Lemma D.1.** *Fix $t \in [m]$. For any picking sequence $\pi^t$ and any valuation profiles $\mathbf{v} = (v_1, \ldots, v_n)$ and $\mathbf{v}^{t,\mathsf{IDO}} = (v_1^{t,\mathsf{IDO}}, \ldots, v_n^{t,\mathsf{IDO}})$ such that for each $i \in N$, $v_i$ and $v_i^{t,\mathsf{IDO}}$ are both consistent with $V_i^t$, we have that*

$$v_i\left(A_i^{\pi^t, \mathbf{v}}\right) \geq v_i^{t,\mathsf{IDO}}\left(A_i^{\pi^t, \mathbf{v}^{t,\mathsf{IDO}}}\right).$$

---

[9] IDO is shorthand for *identical ordering*, as is standard in the literature.

*Proof.* Consider any $t \in [m]$. Suppose we have a picking sequence $\pi^t$ and valuation profiles $\mathbf{v} = (v_1, \ldots, v_n)$ and $\mathbf{v}^{t,\mathsf{IDO}} = (v_1^{t,\mathsf{IDO}}, \ldots, v_n^{t,\mathsf{IDO}})$ such that for each $i \in N$, $v_i$ and $v_i^{t,\mathsf{IDO}}$ are both consistent with $V_i^t$.

An equivalent way to express a picking sequence $\pi^t$ is by looking at the *turns* each agent gets to pick a good. Note that for goods $\{g_t, \ldots, g_m\}$, there will be a total of exactly $m - t + 1$ turns (where at each turn, an agent gets to pick his favorite remaining good).

We introduce several notation:

- Let $\mathsf{turns}_i \subseteq [m - t + 1]$ be the set of turns whereby agents $i$ gets to pick the good under $\pi^t$. Note that $\bigcup_{i \in N} \mathsf{turns}_i = [m - t + 1]$ and for any $i \neq j$, $\mathsf{turns}_i \cap \mathsf{turns}_j = \varnothing$.

- Let $\mathsf{top}_i(k)$ be the $k$-th largest element (counting multiplicity) of the multiset $V_i^t$, for $k \in [m - t + 1]$.

- Let $h_{k,\mathbf{v}}$ be the $k$-th good that was picked according to $\pi^t$ under $\mathbf{v}$.

Consider any agent $i \in N$. Note that for each $k \in \mathsf{turns}_i$ (i.e., in the $k$-th entry of $\pi^t$, it is agent $i$'s turn to pick), under $\mathbf{v}^{t,\mathsf{IDO}}$, since goods in $\{g_t, \ldots, g_m\}$ are identically ordered according to all agents' valuations, agent $i$ will get to (and must) pick her $k$-th favorite good, i.e., $v_i^{t,\mathsf{IDO}}(h_{k,\mathbf{v}^{t,\mathsf{IDO}}}) = \mathsf{top}_i(k)$. Summing over all such $k \in \mathsf{turns}_i$, we get

$$v_i^{t,\mathsf{IDO}}\left(A_i^{\pi^t, \mathbf{v}^{t,\mathsf{IDO}}}\right) = \sum_{k \in \mathsf{turns}_i} v_i^{t,\mathsf{IDO}}(h_{k,\mathbf{v}^{t,\mathsf{IDO}}}) = \sum_{k \in \mathsf{turns}_i} \mathsf{top}_i(k). \tag{17}$$

Now, for each $k \in \mathsf{turns}_i$, under $\mathbf{v}$, we note that agent $i$'s value for the good picked will be at least her value for her $k$-th favorite good, i.e., $v_i(h_{k,\mathbf{v}}) \geq \mathsf{top}_i(k)$ as only $k - 1$ goods have been selected before this turn. Summing over all $k \in \mathsf{turns}_i$, we get

$$v_i\left(A_i^{\pi^t, \mathbf{v}}\right) = \sum_{k \in \mathsf{turns}_i} v_i(h_{k,\mathbf{v}}) \geq \sum_{k \in \mathsf{turns}_i} \mathsf{top}_i(k). \tag{18}$$

Combining (17) and (18), we get $v_i\left(A_i^{\pi^t, \mathbf{v}}\right) \geq v_i^{t,\mathsf{IDO}}\left(A_i^{\pi^t, \mathbf{v}^{t,\mathsf{IDO}}}\right)$, as desired. $\qquad \square$

Next, we introduce the following algorithm (Algorithm 2). Note that given frequency predictions, we know the exact *number* of goods that will arrive (let it be $m$). This allows us to use a **for** loop rather than a **while** loop.

---

**Algorithm 2** Algorithm given frequency predictions and a picking sequence $\pi^1$

---

1: Initialize $A_i \leftarrow \varnothing$ and $V_i^1 \leftarrow V_i$ for all $i \in N$
2: **for** $t \in [m]$ **do**
3:      **for** $i \in N$ **do**
4:          $V_i^{t+1} \leftarrow V_i^t \setminus \{v_i(g_t)\}$
5:          Define $v_i^t$ as follows

$$v_i^t(g_j) := \begin{cases} v_i(g_t) & \text{if } j = t \\ v_i^{t+1,\mathsf{IDO}}(g_j) & \text{otherwise} \end{cases}$$

6:      **end for**
7:      Let $\mathbf{v}^t := (v_1^t, \ldots, v_n^t)$
8:      Let $i^* \in N$ be the agent such that $g_t \in A_{i^*}^{\pi^t, \mathbf{v}^t}$
9:      Let $\pi^{t+1}$ be equivalent to $\pi^t$, except the entry of agent $i^*$ that picked $g_t$ is removed
10:      $A_{i^*} \leftarrow A_{i^*} \cup \{g_t\}$
11: **end for**
12: **return** $\mathcal{A} = (A_1, \ldots, A_n)$

---

Note that simulating the picking sequence on the remaining goods gives a $\mathcal{O}(mn)$ time implementation per round and $\mathcal{O}(m^2 n)$ total time.

Next, we prove the following property about Algorithm 2.

**Lemma D.2.** *Given the picking sequence* $\pi^1$, *Algorithm 2 returns an allocation* $\mathcal{A} = (A_1, \ldots, A_n)$ *such that for every* IDO *valuation profile* $\mathbf{v}^{1,\text{IDO}} = (v_1^{1,\text{IDO}}, \ldots, v_n^{1,\text{IDO}})$ *with* $v_i^{1,\text{IDO}}$ *consistent with* $V_i$ *for all* $i \in N$, *we have for all* $i \in N$,

$$v_i(A_i) \geq v_i^{1,\text{IDO}}\left(A_i^{\pi^1, \mathbf{v}^{1,\text{IDO}}}\right) \quad \text{and} \quad |A_i| = \left|A_i^{\pi^1, \mathbf{v}^{1,\text{IDO}}}\right|.$$

*Proof.* For each $i \in N$ and $t \in [m]$, let $A_i^t$ denote the partial allocation of agent $i$ *after* $g_t$ is allocated to an agent and let $A_i^0 = \varnothing$ be the initial empty allocation. We will prove that for every $i \in N$ and $k \in \{0, \ldots, m\}$,

$$v_i(A_i^k) + v_i^{k+1,\text{IDO}}\left(A_i^{\pi^{k+1}, \mathbf{v}^{k+1,\text{IDO}}}\right) \geq v_i^{1,\text{IDO}}\left(A_i^{\pi^1, \mathbf{v}_i^{1,\text{IDO}}}\right), \text{ and} \tag{19}$$

$$|A_i^k| + \left|A_i^{\pi^{k+1}, \mathbf{v}^{k+1,\text{IDO}}}\right| = \left|A_i^{\pi^1, \mathbf{v}^{1,\text{IDO}}}\right| \tag{20}$$

We proceed by induction. First, consider the base case: we are given that $A_i^0 = \varnothing$ for all $i \in N$. Thus, for all $i \in N$, we have that

(i) $v_i(A_i^0) + v_i^{1,\text{IDO}}\left(A_i^{\pi^1, \mathbf{v}^{1,\text{IDO}}}\right) = v_i(\varnothing) + v_i^{1,\text{IDO}}\left(A_i^{\pi^1, \mathbf{v}^{1,\text{IDO}}}\right) \geq v_i^{1,\text{IDO}}\left(A_i^{\pi^1, \mathbf{v}^{1,\text{IDO}}}\right)$, and

(ii) $|A_i^0| + \left|A_i^{\pi^1, \mathbf{v}^{1,\text{IDO}}}\right| = 0 + \left|A_i^{\pi^1, \mathbf{v}^{1,\text{IDO}}}\right| = \left|A_i^{\pi^1, \mathbf{v}^{1,\text{IDO}}}\right|$,

thus proving the base case where $k = 0$.

Next, suppose that for each $i \in N$ and $\ell \in \{0, \ldots, m-1\}$, we have

$$v_i(A_i^\ell) + v_i^{\ell+1,\text{IDO}}\left(A_i^{\pi^{\ell+1}, \mathbf{v}^{\ell+1,\text{IDO}}}\right) \geq v_i^{1,\text{IDO}}\left(A_i^{\pi^1, \mathbf{v}_i^{1,\text{IDO}}}\right), \text{ and} \tag{21}$$

$$|A_i^\ell| + \left|A_i^{\pi^{\ell+1}, \mathbf{v}^{\ell+1,\text{IDO}}}\right| = \left|A_i^{\pi^1, \mathbf{v}^{1,\text{IDO}}}\right|. \tag{22}$$

We will show that

$$v_i(A_i^{\ell+1}) + v_i^{\ell+2,\text{IDO}}\left(A_i^{\pi^{\ell+2}, \mathbf{v}^{\ell+2,\text{IDO}}}\right) \geq v_i^{1,\text{IDO}}\left(A_i^{\pi^1, \mathbf{v}_i^{1,\text{IDO}}}\right), \text{ and} \tag{23}$$

$$|A_i^{\ell+1}| + \left|A_i^{\pi^{\ell+2}, \mathbf{v}^{\ell+2,\text{IDO}}}\right| = \left|A_i^{\pi^1, \mathbf{v}^{1,\text{IDO}}}\right|. \tag{24}$$

Now, we have that for all $i \in N$ and any $v_i^{\ell+1,\text{IDO}}$, it is easy to observe that

$$\left|A_i^{\pi^{\ell+1}, \mathbf{v}^{\ell+1}}\right| = \left|A_i^{\pi^{\ell+1}, \mathbf{v}^{\ell+1,\text{IDO}}}\right|, \tag{25}$$

since under the same picking sequence $\pi^{\ell+1}$, any agent $i$ should have the same number of goods, regardless of her valuation function.

Let agent $i^* \in N$ be the agent that is allocated $g_{\ell+1}$ by the algorithm. This gives us

$$A_i^{\ell+1} = A_i^\ell \text{ for all } i \in N \setminus \{i^*\}, \text{ and } A_{i^*}^{\ell+1} = A_{i^*}^\ell \cup \{g_{\ell+1}\}. \tag{26}$$

Now, since $\pi^{\ell+2}$ is equivalent to $\pi^{\ell+1}$ but with the entry of agent $i^*$ that picked $g_{\ell+1}$ removed (see Algorithm 2 of the algorithm), we have that

$$A_i^{\pi^{\ell+2}, \mathbf{v}^{\ell+2,\text{IDO}}} = A_i^{\pi^{\ell+1}, \mathbf{v}^{\ell+1}} \text{ for all } i \in N \setminus \{i^*\}, \text{ and } A_{i^*}^{\pi^{\ell+2}, \mathbf{v}^{\ell+2,\text{IDO}}} = A_{i^*}^{\pi^{\ell+1}, \mathbf{v}^{\ell+1}} \setminus \{g_{\ell+1}\}, \tag{27}$$

This holds because in the run of $\pi^{\ell+1}$ under $\mathbf{v}^{\ell+1}$, the good $g_{\ell+1}$ is selected exactly once (by agent $i^*$ at a unique turn); deleting that turn from the picking sequence and deleting $g_{\ell+1}$ from the goods set gives us the same remaining pick process on $\{g_{\ell+2}, \ldots, g_m\}$ (deterministic tie-breaking), and $\mathbf{v}^{\ell+1}$ coincides with $\mathbf{v}^{\ell+2,\text{IDO}}$ on $\{g_{\ell+2}, \ldots, g_m\}$.

Lastly, we note that by Line 5 of the algorithm, for each $i \in N$ and $j \in \{\ell+2, \ldots, m\}$,

$$v_i^{\ell+1}(g_j) = v_i^{\ell+2,\text{IDO}}(g_j). \tag{28}$$

Then, we have that for all agents $i \in N \setminus \{i^*\}$,

$$v_i(A_i^{\ell+1}) + v_i^{\ell+2,\mathsf{IDO}}\left(A_i^{\pi^{\ell+2},\mathbf{v}^{\ell+2,\mathsf{IDO}}}\right) = v_i(A_i^\ell) + v_i^{\ell+2,\mathsf{IDO}}\left(A_i^{\pi^{\ell+2},\mathbf{v}^{\ell+2,\mathsf{IDO}}}\right) \quad \text{(by (26))}$$

$$= v_i(A_i^\ell) + v_i^{\ell+1}\left(A_i^{\pi^{\ell+1},\mathbf{v}^{\ell+1}}\right) \quad \text{(by (27) and (28))}$$

$$\geq v_i(A_i^\ell) + v_i^{\ell+1,\mathsf{IDO}}\left(A_i^{\pi^{\ell+1},\mathbf{v}^{\ell+1,\mathsf{IDO}}}\right) \quad \text{(by Lemma D.1)}$$

$$\geq v_i^{1,\mathsf{IDO}}\left(A_i^{\pi^1,\mathbf{v}^{1,\mathsf{IDO}}}\right) \quad \text{(by (21))}$$

and

$$\left|A_i^{\ell+1}\right| + \left|A_i^{\pi^{\ell+2},\mathbf{v}^{\ell+2,\mathsf{IDO}}}\right| = |A_i^\ell| + \left|A_i^{\pi^{\ell+1},\mathbf{v}^{\ell+1}}\right| \quad \text{(by (26) and (27))}$$

$$= |A_i^\ell| + \left|A_i^{\pi^{\ell+1},\mathbf{v}^{\ell+1,\mathsf{IDO}}}\right| \quad \text{(by (25))}$$

$$= \left|A_i^{\pi^1,\mathbf{v}^{1,\mathsf{IDO}}}\right| \quad \text{(by (22))}.$$

Since $A_{i^*}^{\pi^{\ell+1},\mathbf{v}^{\ell+1}} \setminus \{g_{\ell+1}\} \subseteq \{g_{\ell+2}, \ldots, g_m\}$, (28) implies

$$v_{i^*}^{\ell+2,\mathsf{IDO}}\left(A_{i^*}^{\pi_{\ell+1},\, v^{\ell+1}} \setminus \{g_{\ell+1}\}\right) = v_{i^*}^{\ell+1}\left(A_{i^*}^{\pi^{\ell+1},\mathbf{v}^{\ell+1}} \setminus \{g_{\ell+1}\}\right).$$

Also, by Line 5 of the algorithm, $v_{i^*}^{\ell+1}(g_{\ell+1}) = v_{i^*}(g_{\ell+1})$. By additivity, the sum becomes

$$v_{i^*}(A_{i^*}^\ell) + v_{i^*}^{\ell+1}\left(A_{i^*}^{\pi^{\ell+1},\mathbf{v}^{\ell+1}}\right).$$

Now, we have that

$$v_{i^*}(A_{i^*}^{\ell+1}) + v_{i^*}^{\ell+2,\mathsf{IDO}}\left(A_{i^*}^{\pi^{\ell+2},\mathbf{v}^{\ell+2,\mathsf{IDO}}}\right)$$

$$= v_{i^*}\left(A_{i^*}^\ell \cup \{g_{\ell+1}\}\right) + v_{i^*}^{\ell+2,\mathsf{IDO}}\left(A_{i^*}^{\pi^{\ell+1},\mathbf{v}^{\ell+1}} \setminus \{g_{\ell+1}\}\right) \quad \text{(by (26) and (27))}$$

$$\geq v_{i^*}(A_{i^*}^\ell) + v_{i^*}^{\ell+1,\mathsf{IDO}}\left(A_{i^*}^{\pi^{\ell+1},\mathbf{v}^{\ell+1,\mathsf{IDO}}}\right) \quad \text{(by Lemma D.1)}.$$

and

$$\left|A_{i^*}^{\ell+1}\right| + \left|A_{i^*}^{\pi^{\ell+2},\mathbf{v}^{\ell+2,\mathsf{IDO}}}\right| = \left|A_{i^*}^\ell\right| + \left|A_{i^*}^{\pi^{\ell+1},\mathbf{v}^{\ell+1}}\right| \quad \text{(by (26) and (27))}$$

$$= \left|A_{i^*}^\ell\right| + \left|A_{i^*}^{\pi^{\ell+1},\mathbf{v}^{\ell+1,\mathsf{IDO}}}\right|$$

$$= \left|A_{i^*}^{\pi^1,\mathbf{v}^{1,\mathsf{IDO}}}\right| \quad \text{(by (22))}.$$

Thus, by induction, (19) and (20) holds.

Now, when Algorithm 2 ends (i.e., when $t = m$), we have that for all $i \in N$,

(i) $v_i\left(A_i^m\right) + v_i^{m+1,\mathsf{IDO}}\left(A_i^{\pi^{m+1},\mathbf{v}^{m+1,\mathsf{IDO}}}\right) \geq v_i^{1,\mathsf{IDO}}\left(A_i^{\pi^1,\mathbf{v}_i^{1,\mathsf{IDO}}}\right)$, and

(ii) $|A_i^m| + \left|A_i^{\pi^{m+1},\mathbf{v}^{m+1,\mathsf{IDO}}}\right| = \left|A_i^{\pi^1,\mathbf{v}^{1,\mathsf{IDO}}}\right|$,

where we let $V_i^{m+1} = \varnothing$. Consequently, we get that $A_i^{\pi^{m+1},\mathbf{v}^{m+1,\mathsf{IDO}}} = \varnothing$ for all $i \in N$. This gives us, for all $i \in N$,

$$v_i(A_i) = v_i\left(A_i^m\right) \geq v_i^{1,\mathsf{IDO}}\left(A^{\pi^1,\mathbf{v}_i^{1,\mathsf{IDO}}}\right) \quad \text{and} \quad |A_i| = |A_i^m| = |A_i^{\pi^1,\mathbf{v}^{1,\mathsf{IDO}}}|,$$

as desired, and concludes our proof for this lemma. □

Finally, let $s$ be any share that is feasible in the single-shot fair division setting. This means that for any IDO valuation profile $\mathbf{v}^{1,\mathsf{IDO}} = (v_1^{1,\mathsf{IDO}}, \ldots, v_n^{1,\mathsf{IDO}})$ such that $v_i^{1,\mathsf{IDO}}$ is consistent with $V_i = V_i^1$ for every $i \in N$, there exists an allocation $\mathcal{X} = (X_1, \ldots, X_n)$ such that for all $i \in N$,

$$v_i^{1,\mathsf{IDO}}(X_i) \geq s(V_i, n).$$

Let $\pi$ be the picking sequence such that $\mathcal{X} = \mathcal{A}^{\pi,\mathbf{v}^{1,\mathsf{IDO}}}$. We claim there exists a picking sequence $\pi$ such that $\mathcal{X} = \mathcal{A}^{\pi,\mathbf{v}^{1,\mathsf{IDO}}}$. Since $\mathbf{v}^{1,\mathsf{IDO}}$ is identically ordered, the good chosen at the $k$-th pick is deterministically $g_k$ (ties are broken toward lower-indexed goods). Therefore, defining $\pi$ so that its $k$-th entry equals the agent who receives $g_k$ under $\mathcal{X}$ gives us $\mathcal{A}^{\pi,\mathbf{v}^{1,\mathsf{IDO}}} = \mathcal{X}$.

Then, using $\pi$ and $(V_i)_{i \in N}$ as inputs to Algorithm 2, by Lemma D.2, the algorithm will return an allocation $\mathcal{A} = (A_1, \ldots, A_n)$ such that for all $i \in N$,

$$v_i(A_i) \geq v_i^{1,\mathsf{IDO}}(A_i^{\pi,\mathbf{v}^{1,\mathsf{IDO}}}) = v_i^{1,\mathsf{IDO}}(X_i) \geq s(V_i, n),$$

giving us our result as desired. $\qquad\square$

**Theorem 5.4.** *For $n = 2$, given frequency predictions, there exists an online algorithm that always returns an* EFX *allocation.*

*Proof.* For $i \in \{1, 2\}$ let $T_i := v_i(G) = \sum_{x \in V_i} x$, which is known from the frequency multisets. If $T_i = 0$ for some $i$, then output the allocation that assigns all goods to the other agent. This allocation is EFX since valuations are nonnegative. Hence assume $T_1, T_2 > 0$.

Now rescale each agent's valuation by its total value: define $\widehat{v}_i(\cdot) := v_i(\cdot)/T_i$ for $i \in \{1, 2\}$. Let $\widehat{V}_i$ denote the corresponding (normalized) frequency multiset, i.e., $\widehat{V}_i := \{x/T_i : x \in V_i\}$ (counting multiplicity). Since EFX is invariant under multiplying $v_i$ by a positive constant (the inequality for agent $i$ is scaled on both sides), it suffices to prove the claim for the normalized instance $(\widehat{v}_1, \widehat{v}_2)$ with predictions $(\widehat{V}_1, \widehat{V}_2)$. For readability, we drop hats and assume $v_1(G) = v_2(G) = 1$.

Let $v^{1,\mathsf{IDO}} = (v_1^{1,\mathsf{IDO}}, v_2^{1,\mathsf{IDO}})$ be the IDO valuation profile consistent with $(V_1, V_2)$. Run the *leximin++ cut-and-choose* procedure of Plaut & Roughgarden (2018) on $v^{1,\mathsf{IDO}}$ as follows: agent 1 chooses an (unordered) partition $\{X, Y\}$ of $G$ that minimizes $|v_1^{1,\mathsf{IDO}}(X) - v_1^{1,\mathsf{IDO}}(Y)|$; among minimizers, choose one that maximizes the cardinality of the bundle with smaller $v_1^{1,\mathsf{IDO}}$-value. Agent 2 then chooses her preferred bundle under $v_2^{1,\mathsf{IDO}}$. Let $A' = (A_1', A_2')$ denote the resulting allocation (so $A_2'$ is the bundle chosen by agent 2).

Define a picking sequence $\pi^1$ on $G$ by setting $\pi_k^1 = i$ if and only if $g_k \in A_i'$. Fix the deterministic tie-breaking rule for picking sequences to select the lowest-index good among an agent's most-preferred remaining goods. Since $\mathbf{v}^{1,\mathsf{IDO}}$ is identically ordered, the good selected at the $k$-th pick is $g_k$, and thus defining $\pi^{1,k} = i$ iff $g_k \in A_i'$ gives us $\mathcal{A}_{\pi^1,\mathbf{v}^{1,\mathsf{IDO}}} = A'$.

Run Algorithm 2 on the true online instance using inputs $(V_1, V_2)$ and $\pi^1$, and let $\mathcal{A} = (A_1, A_2)$ be its output. By Lemma D.2, for $i \in \{1, 2\}$ we have

$$v_i(A_i) \geq v_i^{1,\mathsf{IDO}}(A_i') \quad \text{and} \quad |A_i| = |A_i'|. \tag{29}$$

We first show agent 2 is EFX. Since agent 2 chooses her preferred bundle in cut-and-choose,

$$v_2^{1,\mathsf{IDO}}(A_2') \geq \frac{1}{2} v_2^{1,\mathsf{IDO}}(G) = \frac{1}{2}. \tag{30}$$

Combining (29) with (30) gives $v_2(A_2) \geq 1/2$, hence $v_2(A_1) = 1 - v_2(A_2) \leq 1/2$. Therefore, for every $g \in A_1$, we have $v_2(A_2) \geq v_2(A_1) \geq v_2(A_1 \setminus \{g\})$, so agent 2 satisfies EFX.

It remains to show agent 1 is EFX. If $v_1^{1,\mathsf{IDO}}(A_1') \geq v_1^{1,\mathsf{IDO}}(A_2')$, then $v_1^{1,\mathsf{IDO}}(A_1') \geq 1/2$ and by (29) we get $v_1(A_1) \geq 1/2$, which implies $v_1(A_1) \geq v_1(A_2) \geq v_1(A_2 \setminus \{g\})$ for all $g \in A_2$. Thus, assume

$$v_1^{1,\mathsf{IDO}}(A_1') < v_1^{1,\mathsf{IDO}}(A_2'). \tag{31}$$

Let $a := v_1^{1,\mathsf{IDO}}(A_1')$, so $a < 1/2$ and $v_1^{1,\mathsf{IDO}}(A_2') = 1 - a$.

Suppose for contradiction that agent 1 violates EFX in $\mathcal{A}$. Then there exists some $g \in A_2$ with $v_1(A_1) < v_1(A_2 \setminus \{g\})$. Let $g^* \in \arg\min_{h \in A_2} v_1(h)$; since removing the least-valued good leaves the largest remainder, we also have

$$v_1(A_1) < v_1(A_2 \setminus \{g^*\}). \tag{32}$$

Because $v_1$ and $v_1^{1,\text{IDO}}$ are both consistent with $V_1$, there exists a bijection $f : G \to G$ such that $v_1(g) = v_1^{1,\text{IDO}}(f(g))$ for all $g \in G$. Define

$$B_2 := \{f(g) \mid g \in A_2 \setminus \{g^*\}\}, \quad B_1 := G \setminus B_2.$$

Then $\mathcal{B} = (B_1, B_2)$ is a partition of $G$ and, by construction,

$$v_1^{1,\text{IDO}}(B_2) = v_1(A_2 \setminus \{g^*\}).$$

Let $x := v_1(A_1)$ and $\delta := v_1(g^*) \geq 0$. Since $v_1(G) = 1$ and $A_1 \cup A_2 = G$, we have $v_1(A_2 \setminus \{g^*\}) = 1 - x - \delta$. By (32), $1 - x - \delta > x$, and by (29), $x \geq a$. Thus,

$$v_1^{1,\text{IDO}}(B_2) = 1 - x - \delta > x \geq a.$$

Also,

$$v_1^{1,\text{IDO}}(B_2) = 1 - x - \delta \leq 1 - a,$$

with equality only if $x = a$ and $\delta = 0$. If the inequality is strict, then $v_1^{1,\text{IDO}}(B_2) \in (a, 1 - a)$. Since $a < 1/2$ and $v_1^{1,\text{IDO}}(B_2) \in (a, 1 - a)$, we have $|1 - 2v_1^{1,\text{IDO}}(B_2)| < |1 - 2a|$. Thus,

$$|v_1^{1,\text{IDO}}(B_1) - v_1^{1,\text{IDO}}(B_2)| = |1 - 2v_1^{1,\text{IDO}}(B_2)| < |1 - 2a| = |v_1^{1,\text{IDO}}(A_1') - v_1^{1,\text{IDO}}(A_2')|,$$

contradicting the choice of $\{A_1', A_2'\}$ as a minimizer of $|v_1^{1,\text{IDO}}(X) - v_1^{1,\text{IDO}}(Y)|$.

In the remaining case, we have $v_1^{1,\text{IDO}}(B_2) = 1 - a$ and $v_1^{1,\text{IDO}}(B_1) = a$, so $\mathcal{B}$ is also a minimizer of the absolute difference. Moreover, $|B_1| = |A_1| + 1 = |A_1'| + 1$ (using (29)), and $B_1$ is the smaller-valued bundle for agent 1 (because $a < 1 - a$). This contradicts the tie-breaking rule that, among difference-minimizing partitions, maximizes the cardinality of the smaller-valued bundle.

Therefore, (32) is impossible and agent 1 satisfies EFX. Thus, $\mathcal{A}$ is EFX. $\qquad\square$

**Theorem 5.5.** *For $n \geq 3$, given frequency predictions, there does not exist any online algorithm with a competitive ratio strictly larger than $0$ with respect to* EFX.

*Proof.* Suppose for a contradiction that there exists a $\gamma$-competitive algorithm for approximating EFX when $n \geq 3$ and with frequency predictions, for some $\gamma \in (0, 1]$. Let $\varepsilon > 0$ be a small constant and $K > 0$ be sufficiently large. Let $V_i = \{K^2, \ldots, K^2, K, \varepsilon, \varepsilon\}$ be the multiset with $n + 1$ elements for each $i \in N$. Let the $n + 1$ goods be $g_1, \ldots, g_{n+1}$ with valuations as follows:

| $\mathbf{v}$ | $g_1$ | $g_2$ | $g_3$ | $g_4$ | $\cdots$ | $g_n$ | $g_{n+1}$ |
|---|---|---|---|---|---|---|---|
| 1 | $K$ | $K^2$ | $\varepsilon$ | $K^2$ | $\ldots$ | $K^2$ | $\varepsilon$ |
| 2 | $K$ | $\varepsilon$ | $\varepsilon$ | $K^2$ | $\ldots$ | $K^2$ | $K^2$ |
| $\vdots$ | $\vdots$ | $\vdots$ | $\vdots$ | $\vdots$ | $\ldots$ | $\vdots$ | $\vdots$ |
| $n$ | $K$ | $\varepsilon$ | $\varepsilon$ | $K^2$ | $\ldots$ | $K^2$ | $K^2$ |

Suppose w.l.o.g. that $g_1$ is allocated to agent $1$. We split our analysis into two cases.

**Case 1: $g_2$ is allocated to agent $1$.** Then no goods in $\{g_3, \ldots, g_{n+1}\}$ can be allocated to agent $1$ (otherwise there must exist some agent in $N \setminus \{1\}$ that receives no goods, leading to a $0$-EFX allocation). Moreover, if some agent in $N \setminus \{1\}$ receives two goods from $\{g_3, \ldots, g_{n+1}\}$, then there must exist some agent other agent from $N \setminus \{1\}$ that receives no goods, leading to a $0$-EFX allocation. Thus, we must have that each good in $\{g_3, \ldots, g_{n+1}\}$ is allocated to a different agent in $N \setminus \{1\}$. Let the agent that is allocated $g_3$ be $i$. Then, $v_i(A_i) = \varepsilon$ and $v_i(A_1 \setminus \{g_2\}) = K$. This gives us $\gamma \leq \frac{\varepsilon}{K} \to 0$ as $K \to \infty$, a contradiction.

**Case 2:** $g_2$ **is not allocated to agent** $1$**.** Suppose w.l.o.g. that $g_2$ is allocated to agent $2$ instead. If at least one of $\{g_3, \ldots, g_{n+1}\}$ is allocated to agent $2$ and agent $1$ is not allocated any good from $\{g_4, \ldots, g_n\}$, then $v_1(A_1) \leq K + 2\varepsilon$ and $v_1(A_2 \setminus \{g\}) \geq K^2$ for $g \in \arg\min_{g' \in A_2} v_1(g')$. This gives us

$$\gamma \leq \frac{K + 2\varepsilon}{K^2} = \frac{1}{K} + \frac{2\varepsilon}{K^2} \to 0$$

as $K \to \infty$, a contradiction. Therefore, if at least one of $\{g_3, \ldots, g_{n+1}\}$ is allocated to agent $2$, agent $1$ must be allocated some good from $\{g_4, \ldots, g_n\}$. However, this means that there are at most $n - 3$ remaining goods and $n - 2$ agents, leaving some agent in $N \setminus \{1, 2\}$ with an empty bundle, leading to a $0$-EFX allocation. Thus, none of the goods in $\{g_3, \ldots, g_{n+1}\}$ can be allocated to agent $2$. This implies that there must be some agent $i \in N \setminus \{2\}$ that receives two goods eventually. However, $v_2(A_2) = \varepsilon$ and $v_2(A_i \setminus \{g\}) \geq K$ for $g \in \arg\min_{g' \in A_i} v_2(g')$. This gives us $\gamma \leq \frac{\varepsilon}{K} \to 0$ as $K \to \infty$, a contradiction.

We arrived at a contradiction in both cases, thereby proving our claim. $\square$

### D.1. Noisy Frequency Predictions: Instantiation Model and Proof of Theorem 5.6

We fix predicted frequency multisets $(\widehat{V}_i)_{i \in N}$. As in the frequency prediction model, we assume the number of arriving goods is $m$ and that $|\widehat{V}_i| = m$ for every $i \in N$.

**Online instantiation and virtual valuations.** Fix an agent $i \in N$ and a predicted multiset $\widehat{V}_i$. An *online instantiation rule* maintains a multiset of unused predicted values $(\widehat{V}_i^{\,t})_{t \in [m+1]}$, initialized as $\widehat{V}_i^{\,1} := \widehat{V}_i$. Upon the arrival of good $g_t$ (for $t \in [m]$), it selects a value $\widetilde{v}_i(g_t) \in \widehat{V}_i^{\,t}$ and updates $\widehat{V}_i^{\,t+1} := \widehat{V}_i^{\,t} \setminus \{\widetilde{v}_i(g_t)\}$, where $\setminus$ denotes *multiset difference* (i.e., removing a single occurrence). This defines a virtual additive valuation $\widetilde{v}_i$ on $G$; in particular,

$$\{\widetilde{v}_i(g_1), \ldots, \widetilde{v}_i(g_m)\} = \widehat{V}_i \quad \text{as multisets.}$$

**Error measures.** The (agent specific) absolute instantiation error is

$$\eta_i := \sum_{t=1}^{m} \left| v_i(g_t) - \widetilde{v}_i(g_t) \right|.$$

To align with multiplicative guarantees, define the relative error

$$\varepsilon_i := \begin{cases} 0, & \text{if } s(\widehat{V}_i, n) = 0, \\ \min\{1, \ \eta_i / s(\widehat{V}_i, n)\}, & \text{if } s(\widehat{V}_i, n) > 0. \end{cases}$$

**Interpreting $\eta_i$ via multiset distances.** For two multisets $V$ and $\widehat{V}$ of equal cardinality, define the optimal matching (earthmover) cost

$$\mathrm{EMD}_1(V, \widehat{V}) := \min_{\sigma: \widehat{V} \to V \text{ bijection}} \sum_{x \in \widehat{V}} \left| x - \sigma(x) \right|.$$

This quantity is the 1-Wasserstein distance between the corresponding empirical distributions (up to normalization). In our instantiation model, any online instantiation rule induces a bijection between predicted values and realized values, and hence its absolute instantiation error satisfies $\eta_i \geq \mathrm{EMD}_1(V_i, \widehat{V}_i)$. Thus, $\eta_i$ should be viewed as an *end-to-end* error parameter capturing both (i) prediction quality and (ii) the online instantiation procedure's ability to match predictions to the realized stream order.

**Algorithm.** Maintain an online instantiation rule for each agent $i$. On each arrival $g_t$, first instantiate the virtual values $(\widetilde{v}_i(g_t))_{i \in N}$ and update the unused multisets $(\widehat{V}_i^{\,t})_{i \in N}$. Then allocate $g_t$ by running the online algorithm guaranteed by Theorem 5.1 on the *virtual* instance with valuations $(\widetilde{v}_i)_{i \in N}$ and frequency predictions $(\widehat{V}_i)_{i \in N}$.

**Theorem 5.6.** *Let $s$ be a feasible share. There exists an online algorithm that, given the predicted multisets $(\widehat{V}_i)_{i \in N}$, outputs an allocation $\mathcal{A}$ such that for every agent $i \in N$, $v_i(A_i) \geq (1 - \varepsilon_i) \cdot s(\widehat{V}_i, n)$, where $\varepsilon_i \in [0, 1]$ is the relative instantiation error induced by the algorithm's online instantiation rule.*

*Proof of Theorem 5.6.* Fix the predicted multisets $(\widehat{V}_i)_{i\in N}$. The algorithm maintains, for each agent $i$, an online instantiation rule, inducing a virtual valuation $\widetilde{v}_i$ and an error $\eta_i$ as defined above. Let $\mathcal{A} = (A_1, \ldots, A_n)$ be the final allocation returned by running the online algorithm from Theorem 5.1 on the virtual instance $(N, G, (\widetilde{v}_i)_{i\in N})$ with frequency predictions $(\widehat{V}_i)_{i\in N}$.

By construction, for each agent $i$ the frequency multiset of $\widetilde{v}_i$ equals $\widehat{V}_i$. Therefore, applying Theorem 5.1 to the virtual instance gives us

$$\widetilde{v}_i(A_i) \ \geq \ s(\widehat{V}_i, n) \qquad \text{for all } i \in N. \tag{33}$$

Now fix any agent $i \in N$. Using the inequality $x \geq y - |x - y|$ (valid for all real $x, y$), we have for each good $g$

$$v_i(g) \ \geq \ \widetilde{v}_i(g) - \big|v_i(g) - \widetilde{v}_i(g)\big|.$$

Summing over the goods in $A_i$ and using additivity gives us

$$v_i(A_i) = \sum_{g \in A_i} v_i(g) \geq \sum_{g \in A_i} \widetilde{v}_i(g) - \sum_{g \in A_i} \big|v_i(g) - \widetilde{v}_i(g)\big| = \widetilde{v}_i(A_i) - \sum_{g \in A_i} \big|v_i(g) - \widetilde{v}_i(g)\big|.$$

Since $A_i \subseteq G$, we have $\sum_{g \in A_i} |v_i(g) - \widetilde{v}_i(g)| \leq \sum_{t=1}^{m} |v_i(g_t) - \widetilde{v}_i(g_t)| = \eta_i$, and thus, together with (33),

$$v_i(A_i) \ \geq \ s(\widehat{V}_i, n) - \eta_i. \tag{34}$$

If $s(\widehat{V}_i, n) = 0$, then $\varepsilon_i = 0$ and the desired inequality $v_i(A_i) \geq (1 - \varepsilon_i)s(\widehat{V}_i, n)$ holds trivially. Assume $s(\widehat{V}_i, n) > 0$. If $\eta_i \geq s(\widehat{V}_i, n)$, then $\varepsilon_i = 1$ and the right-hand side is 0; since valuations are nonnegative, $v_i(A_i) \geq 0$ and the claim follows. Otherwise $\eta_i < s(\widehat{V}_i, n)$, so $\varepsilon_i = \eta_i/s(\widehat{V}_i, n)$ and (34) implies

$$v_i(A_i) \ \geq \ s(\widehat{V}_i, n) - \eta_i = \left(1 - \frac{\eta_i}{s(\widehat{V}_i, n)}\right) s(\widehat{V}_i, n) = (1 - \varepsilon_i)\, s(\widehat{V}_i, n).$$

This holds for every agent $i \in N$, completing the proof. $\qquad\square$

**A simple instantiation rule.** A concrete way to instantiate predicted multisets online is: for each agent $i$, maintain the multiset of unused predicted values, and upon arrival of good $g_t$ match $v_i(g_t)$ to the closest unused predicted value (breaking ties arbitrarily), setting $\widetilde{v}_i(g_t)$ to that value. This rule can be implemented in $\mathcal{O}(\log m)$ time per agent per round using a balanced binary search tree. Our guarantees in Theorem 5.6 apply to any instantiation rule; analyzing instantiation rules that bound $\eta_i$ in terms of a distance between multisets is an interesting direction.

# E. Section 6: Identical Valuations

Finally, we also study the special case where all agents have *identical valuation functions*. This setting is motivated by both theoretical and practical considerations. From a theoretical perspective, this case serves as a natural benchmark, and aligns with a rich line of work in fair division that has focused on this restricted but structurally appealing setting (Barman & Sundaram, 2020; Elkind et al., 2025a; Mutzari et al., 2023; Plaut & Roughgarden, 2018). From an applied perspective, this setting captures scenarios where agents value goods based on uniform or shared criteria (e.g., identical machines processing uniform jobs), and is especially relevant in the context of online multiprocessor scheduling. In particular, a large body of work on semi-online scheduling (Cheng et al., 2005; Kellerer et al., 1997) considers the goal of minimizing the makespan—the maximum load assigned to any processor—when tasks arrive sequentially (analogous to MMS in our setting). The majority of these models (and other variants in the online scheduling literature) assume identical valuations (i.e., identical machines) (refer to the survey by Dwibedy & Mohanty (2022)). In particular, it is known that in this setting, one can achieve $\frac{2}{3}$-MMS for $n = 2$ (Kellerer et al., 1997) and $\frac{1}{n-1}$-MMS for $n \geq 3$ (Tan & Wu, 2007).

We further study the fairness guarantees achievable in this setting beyond MMS. Observe that, under identical valuations with frequency predictions, the problem collapses to the classic single-shot offline fair-division setting—so any offline guarantee (for example, the existence of EFX allocations for any number of agents) immediately applies here. Consequently, we focus on the two remaining settings: no information and normalization information. Importantly, even though each agent's valuation for any given good is the same, the precise value of each future arrival remains unknown.

In the single-shot setting, it is known that EFX allocations always exist (and can be computed in polynomial time) under identical valuations for any number of agents (Plaut & Roughgarden, 2018). On the positive side, the algorithm of Elkind et al. (2025a, Theorem 3.7) (which allocates the next arriving good to an agent with the least bundle value so far) shows that EF1 allocations can be found for identical valuations in the online setting no future information, which we reproduce here for completeness.

**Proposition E.1** (Elkind et al. (2025a), Theorem 3.7). *For $n \geq 2$, under identical valuations and without future information, there exists an online algorithm that always returns an EF1 allocation.*

We complement these known positive results with an impossibility result for EFX. Thus, even in this setting with identical valuations, the lack of future information can still limit fairness guarantees.

**Proposition E.2.** *For $n \geq 2$, under identical valuations and without future information, there does not exist any online algorithm with a competitive ratio strictly larger than 0 with respect to EFX.*

*Proof.* Suppose for a contradiction that there exists a $\gamma$-competitive algorithm for approximating EFX under identical valuations, for some $\gamma \in (0, 1]$. Fix $K > 1/\gamma$. Consider the case of $n = 2$, and let the first two goods be $g_1$ and $g_2$ with $v(g_1) = v(g_2) = 1$. Let agent 1 denote the recipient of $g_1$ (renaming agents if necessary). If $g_2$ is also allocated to agent 1, let the third (and final) good be $g_3$ with $v(g_3) = 0$. We have $v(A_2) = 0$ and $g_2 \in A_1$, so $v(A_1 \setminus \{g_1\}) \geq v(g_2) = 1$. Thus, the allocation is not $\gamma$-EFX for any $\gamma > 0$ (consider the pair $(2, 1)$ and the good $g_1 \in A_1$).

Thus, $g_2$ has to be allocated to agent 2. Then, let the third (and final) good be $g_3$ with $v(g_3) = K$. If $g_3$ is allocated to agent 1, then for the pair $(2, 1)$ and the good $g_1 \in A_1$ we must have

$$v(A_2) = 1 \geq \gamma \cdot v(A_1 \setminus \{g_1\}) = \gamma K,$$

so $\gamma \leq 1/K$. If instead $g_3$ is allocated to agent 2, then for the pair $(1, 2)$ and the good $g_2 \in A_2$ we must have

$$v(A_1) = 1 \geq \gamma \cdot v(A_2 \setminus \{g_2\}) = \gamma K,$$

so again $\gamma \leq 1/K$. This implies $\gamma \leq 1/K < \gamma$, a contradiction.

For $n \geq 3$, this follows immediately from Theorem E.4: if a $\gamma$-competitive algorithm existed without future information, then the same algorithm (ignoring any advice) would also be a $\gamma$-competitive algorithm in the stronger model where normalization information is provided, contradicting Theorem E.4. □

Theorem 4.3 and Proposition E.2 essentially give strong impossibility results for achieving approximate EFX even in the case of two agents, even if we assume either normalization information or identical valuations, respectively. In what follows, we consider the setting where we have *both* normalization information and identical valuations. For two agents, we show that while an EFX allocation still may fail to exist, we can obtain a $\frac{\sqrt{5}-1}{2}$-EFX allocation, and also prove that this bound is tight (i.e., no online algorithm can do better than this given the assumptions).

**Theorem E.3.** *For $n = 2$, under identical valuations and given normalization information, there exists an online algorithm that returns a $\frac{\sqrt{5}-1}{2}$-EFX allocation. Moreover, in this setting, there does not exist any online algorithm with a competitive ratio strictly larger than $\frac{\sqrt{5}-1}{2}$ with respect to EFX.*

*Proof.* We first show the upper bound. Suppose for a contradiction that there exists a $\gamma$-competitive algorithm for approximating EFX when $n = 2$, for some $\gamma > \frac{\sqrt{5}-1}{2}$. Let $k = \lceil \frac{\sqrt{5}-2}{\varepsilon} \rceil$, and let the first $k$ goods be $g_1, \ldots, g_k$ with values $v(g_1) = \cdots = v(g_k) = \varepsilon$. If each agent is allocated at least one of these $k$ goods, let the $(k + 1)$-th (and final) good be $g_{k+1}$ with $v(g_{k+1}) = 1 - k\varepsilon$. Suppose $g_{k+1}$ is allocated to agent $i \in \{1, 2\}$, and let the other agent be $j$. We have $v(A_j) \leq (k - 1)\varepsilon < \sqrt{5} - 2$. Moreover, for $\varepsilon$ sufficiently small we have $v(g_{k+1}) = 1 - k\varepsilon > \varepsilon$, so any $g^* \in \arg\min_{g' \in A_i} v(g')$ satisfies $v(g^*) = \varepsilon$ and thus

$$v(A_i \setminus \{g^*\}) \geq v(g_{k+1}) = 1 - k\varepsilon.$$

Consequently, since the allocation must be $\gamma$-EFX, for $g^* \in \arg\min_{g' \in A_i} v(g')$ we have

$$v(A_j) \geq \gamma \cdot v(A_i \setminus \{g^*\}),$$

and thus

$$\gamma \le \frac{v(A_j)}{v(A_i \setminus \{g^*\})} \le \frac{(k-1)\varepsilon}{1-k\varepsilon}.$$

Moreover, $k = \left\lceil \frac{\sqrt{5}-2}{\varepsilon} \right\rceil$ implies $(k-1)\varepsilon < \sqrt{5} - 2$ and $k\varepsilon \to \sqrt{5} - 2$ as $\varepsilon \to 0$, so the right-hand side converges to

$$\frac{\sqrt{5}-2}{3-\sqrt{5}} = \frac{\sqrt{5}-1}{4} < \frac{\sqrt{5}-1}{2}.$$

Thus, for $\varepsilon > 0$ sufficiently small we obtain $\gamma < \frac{\sqrt{5}-1}{2}$, contradicting the assumption that $\gamma > \frac{\sqrt{5}-1}{2}$. Thus, all $k$ goods must be allocated to a single agent. W.l.o.g., let it be agent 1. Then, let the next two (and final) goods be $g_{k+1}$ and $g_{k+2}$ with $v(g_{k+1}) = v(g_{k+2}) = \frac{1-k\varepsilon}{2}$. If $g_{k+1}$ is allocated to agent 1, then $v(A_2) \le \frac{1-k\varepsilon}{2}$ and $v(A_1 \setminus \{g\}) \ge (k-1) \cdot \varepsilon + \frac{1-k\varepsilon}{2}$ where $g \in \arg\min_{g' \in A_1} v(g')$. We get that

$$\gamma \le \frac{\frac{1-k\varepsilon}{2}}{(k-1) \cdot \varepsilon + \frac{1-k\varepsilon}{2}} = \frac{1-k\varepsilon}{k\varepsilon - 2\varepsilon + 1}. \tag{35}$$

Since $\frac{\sqrt{5}-2}{\varepsilon} \le k < \frac{\sqrt{5}-2}{\varepsilon} + 1$, multiplying by $\varepsilon$ throughout gives us

$$\sqrt{5} - 2 \le k\varepsilon \le \sqrt{5} - 2 + \varepsilon.$$

By the squeeze theorem, we get $\lim_{\varepsilon \to 0} k\varepsilon = \sqrt{5} - 2$. Let

$$f(\varepsilon) := \frac{1-k\varepsilon}{k\varepsilon - 2\varepsilon + 1}.$$

The $\gamma$-EFX constraint in this subcase implies $\gamma \le f(\varepsilon)$. Since $k\varepsilon \to \sqrt{5} - 2$ as $\varepsilon \to 0$, we have

$$\lim_{\varepsilon \to 0} f(\varepsilon) = \frac{\sqrt{5}-1}{2}.$$

Fix $\varepsilon > 0$ sufficiently small so that

$$f(\varepsilon) < \gamma \qquad \text{and} \qquad \frac{2k\varepsilon}{1-k\varepsilon} < \gamma.$$

Such an $\varepsilon$ exists since both $f(\varepsilon)$ and $\frac{2k\varepsilon}{1-k\varepsilon}$ converge to $\frac{\sqrt{5}-1}{2} < \gamma$ as $\varepsilon \to 0$. Then $\gamma \le f(\varepsilon) < \gamma$, a contradiction.

Thus, suppose $g_{k+1}$ is allocated to agent 2. Then, if both final goods go to agent 2, $v(A_1) = k\varepsilon$ and $A_2$ contains two goods each of value $(1-k\varepsilon)/2$. For $\gamma$-EFX, considering agent 1 vs. agent 2 and removing one of those equal goods from $A_2$,

$$v(A_1) \ge \gamma v(A_2 \setminus \{g\}) = \gamma \cdot \frac{1-k\varepsilon}{2},$$

so $\gamma \le \frac{2k\varepsilon}{1-k\varepsilon}$. In this subcase, $\gamma$-EFX implies

$$\gamma \le \frac{2k\varepsilon}{1-k\varepsilon}.$$

As $k\varepsilon \to \sqrt{5} - 2$ when $\varepsilon \to 0$, the right-hand side converges to $\frac{\sqrt{5}-1}{2}$. By our choice of $\varepsilon$, we have $\frac{2k\varepsilon}{1-k\varepsilon} < \gamma$, which contradicts $\gamma \le \frac{2k\varepsilon}{1-k\varepsilon}$.

Now, suppose agent 2 gets exactly one of the two final goods. Then $v(A_2) = \frac{1-k\varepsilon}{2}$ and agent 1's bundle contains $k$ goods of value $\varepsilon$ and one good of value $\frac{1-k\varepsilon}{2}$. Let $g \in \arg\min_{g' \in A_1} v(g')$. Then

$$v(A_1 \setminus \{g\}) \ge (k-1)\varepsilon + \frac{1-k\varepsilon}{2}.$$

Since the allocation must be $\gamma$-EFX, for the ordered pair $(2, 1)$ we have $v(A_2) \ge \gamma \cdot v(A_1 \setminus \{g\})$, and hence

$$\gamma \le \frac{v(A_2)}{v(A_1 \setminus \{g\})} \le \frac{\frac{1-k\varepsilon}{2}}{(k-1)\varepsilon + \frac{1-k\varepsilon}{2}} = \frac{1-k\varepsilon}{k\varepsilon - 2\varepsilon + 1},$$

---

**Algorithm 3** Returns a $\frac{(\sqrt{5}-1)}{2}$-EFX allocation for $n = 2$, given normalization information, and agents have identical valuations

---

1: Initialize $\mathcal{A} = (A_1, A_2) = (\varnothing, \varnothing)$
2: **while** there exists a good $g$ that arrives **do**
3:    **if** $v(A_1 \cup \{g\}) \leq \frac{(\sqrt{5}-1)}{2}$ **then**
4:       $A_1 \leftarrow A_1 \cup \{g\}$
5:    **else**
6:       $A_2 \leftarrow A_2 \cup \{g\}$
7:    **end if**
8: **end while**
9: **return** $\mathcal{A} = (A_1, A_2)$

---

contradicting the choice of $\varepsilon$.

Next, we prove the lower bound. Consider the allocation $\mathcal{A}$ returned by the following algorithm (Algorithm 3).

We have that

$$v(A_2) \geq 1 - \frac{\sqrt{5}-1}{2} = \frac{3 - \sqrt{5}}{2} \quad \text{and} \quad v(A_1) \leq \frac{\sqrt{5}-1}{2}.$$

Then, we get that

$$v(A_2) \geq \frac{3 - \sqrt{5}}{2} = \left(\frac{\sqrt{5}-1}{2}\right)^2 \geq \frac{\sqrt{5}-1}{2} \cdot v(A_1). \tag{36}$$

Fix any $g \in A_2$, and let $A_1^{\text{pre}}$ denote agent 1's bundle immediately before $g$ arrived. Then, we have that

$$v(A_1^{\text{pre}} \cup \{g\}) > \frac{\sqrt{5}-1}{2},$$

so

$$v(g) > \frac{\sqrt{5}-1}{2} - v(A_1^{\text{pre}}).$$

Because $v(A_1)$ is monotone nondecreasing over the run of the algorithm, we have $v(A_1^{\text{pre}}) \leq v(A_1)$ at termination, and therefore

$$v(g) > \frac{\sqrt{5}-1}{2} - v(A_1).$$

We split our analysis into two cases.

**Case 1:** $v(A_1) < \sqrt{5} - 2$**.** Then, algebraic manipulation gives us

$$v(A_2) = 1 - v(A_1) < 2 \times \left(\frac{\sqrt{5}-1}{2} - v(A_1)\right) < 2 \times \min_{g \in A_2} v(g).$$

This means that $|A_2| = 1$ and $v(A_1) \geq 0 = v(A_2 \setminus \{g\})$ for any $g \in A_2$. Together with (36), the result follows.

**Case 2:** $v(A_1) \geq \sqrt{5} - 2$**.** If $v(A_2 \setminus \{g\}) = 0$ for some $g \in A_2$, then $v(A_1) \geq \frac{\sqrt{5}-1}{2} \cdot v(A_2 \setminus \{g\})$ holds trivially. Otherwise, fix any $g \in A_2$ with $v(A_2 \setminus \{g\}) > 0$. By normalization, $v(A_2) = 1 - v(A_1)$, and we have $v(g) > \frac{\sqrt{5}-1}{2} - v(A_1)$. Thus,

$$v(A_2 \setminus \{g\}) = v(A_2) - v(g) < (1 - v(A_1)) - \left(\frac{\sqrt{5}-1}{2} - v(A_1)\right) = 1 - \frac{\sqrt{5}-1}{2}.$$

Therefore,

$$\frac{v(A_1)}{v(A_2 \setminus \{g\})} > \frac{v(A_1)}{1 - \frac{\sqrt{5}-1}{2}} \geq \frac{\sqrt{5}-2}{1 - \frac{\sqrt{5}-1}{2}} = \frac{\sqrt{5}-1}{2},$$

where the inequality uses the assumption of this case that $v(A_1) \geq \sqrt{5} - 2$. This shows that for every $g \in A_2$ we have $v(A_1) \geq \frac{\sqrt{5}-1}{2} \cdot v(A_2 \setminus \{g\})$. On the other hand, (36) implies $v(A_2) \geq \frac{\sqrt{5}-1}{2} \cdot v(A_1)$, and since $v(A_1 \setminus \{g\}) \leq v(A_1)$ for all $g \in A_1$, we also have $v(A_2) \geq \frac{\sqrt{5}-1}{2} \cdot v(A_1 \setminus \{g\})$ for all $g \in A_1$. Thus, the allocation is $\frac{\sqrt{5}-1}{2}$-EFX.

Thus, the algorithm gives us a $\frac{\sqrt{5}-1}{2}$-EFX allocation, and a lower bound in this setting. $\qquad\square$

However, the strong impossibility result still persists for $n \geq 3$.

**Theorem E.4.** *For $n \geq 3$, under identical valuations and given normalization information, there does not exist any online algorithm with a competitive ratio strictly larger than $0$ with respect to* EFX.

*Proof.* Suppose for a contradiction that there exists an online algorithm that is $\gamma$-competitive for approximating EFX when $n \geq 3$, under identical valuations and given normalization information, for some $\gamma \in (0, 1]$. Let $\varepsilon > 0$ be specified later, and recall that normalization means $v(G) = 1$.

Consider an instance with $m := n + 1$ goods. The first two goods are $g_1, g_2$ with $v(g_1) = v(g_2) = \varepsilon$. Let agent 1 be the recipient of $g_1$ (renaming agents if necessary).

**Case 1:** $g_2$ is allocated to agent 1. Let $g_3, \ldots, g_{n+1}$ arrive with

$$v(g_3) = 1 - 2\varepsilon \qquad \text{and} \qquad v(g_4) = \cdots = v(g_{n+1}) = 0,$$

so that $\sum_{t=1}^{n+1} v(g_t) = 1$. At most two agents can receive positive total value (agent 1, and possibly the recipient of $g_3$), so there exists an agent $i \in N \setminus \{1\}$ with $v(A_i) = 0$. Since $g_1 \in A_1$ and $g_2 \in A_1$, we have $v(A_1 \setminus \{g_1\}) \geq v(g_2) = \varepsilon$. The $\gamma$-EFX condition for the ordered pair $(i, 1)$ and the good $g_1 \in A_1$ requires

$$0 = v(A_i) \geq \gamma \cdot v(A_1 \setminus \{g_1\}) \geq \gamma\varepsilon,$$

a contradiction for any $\gamma > 0$.

**Case 2:** $g_2$ is not allocated to agent 1. Let $g_3, \ldots, g_{n+1}$ arrive with

$$v(g_k) = \frac{1 - 2\varepsilon}{n - 1} \qquad \text{for all } k \in \{3, \ldots, n+1\},$$

so again $\sum_{t=1}^{n+1} v(g_t) = 1$. Among these $n - 1$ goods and $n$ agents, there exists an agent $i$ who receives no good from $\{g_3, \ldots, g_{n+1}\}$; hence $v(A_i) \leq \varepsilon$ (since every agent receives at most one of $g_1, g_2$). Because $m = n + 1$, by the pigeonhole principle there exists an agent $j$ with $|A_j| \geq 2$. Moreover, since $g_1$ and $g_2$ are allocated to two different agents, agent $j$ must receive at least one good from $\{g_3, \ldots, g_{n+1}\}$, and therefore $j \neq i$.

Let $g^* \in \arg\min_{g \in A_j} v(g)$. Since $A_j$ contains at least one good from $\{g_3, \ldots, g_{n+1}\}$ and each such good has value $\frac{1-2\varepsilon}{n-1}$, we claim that

$$v(A_j \setminus \{g^*\}) \geq \frac{1 - 2\varepsilon}{n - 1}.$$

Indeed, if $g^* \notin \{g_3, \ldots, g_{n+1}\}$ then $A_j \setminus \{g^*\}$ still contains a good from $\{g_3, \ldots, g_{n+1}\}$, so the inequality holds. Otherwise $g^* \in \{g_3, \ldots, g_{n+1}\}$, and by minimality every good in $A_j \setminus \{g^*\}$ has value at least $v(g^*) = \frac{1-2\varepsilon}{n-1}$; since $|A_j| \geq 2$, the set $A_j \setminus \{g^*\}$ is nonempty, so the inequality again holds.

Applying $\gamma$-EFX to the ordered pair $(i, j)$ and the good $g^* \in A_j$ gives

$$v(A_i) \geq \gamma \cdot v(A_j \setminus \{g^*\}) \quad \Rightarrow \quad \gamma \leq \frac{v(A_i)}{v(A_j \setminus \{g^*\})} \leq \frac{\varepsilon}{(1 - 2\varepsilon)/(n-1)} = \frac{(n-1)\varepsilon}{1 - 2\varepsilon}.$$

Choose

$$\varepsilon \in \left(0, \ \min\left\{\frac{\gamma}{2(n-1)}, \ \frac{1}{4}\right\}\right).$$

Then $1 - 2\varepsilon > \frac{1}{2}$, and hence

$$\frac{(n-1)\varepsilon}{1 - 2\varepsilon} < \frac{(n-1) \cdot \gamma/(2(n-1))}{1/2} = \gamma,$$

contradicting $\gamma \leq \frac{(n-1)\varepsilon}{1-2\varepsilon}$.

In both cases we obtain a contradiction; therefore no online algorithm can have competitive ratio strictly larger than 0 with respect to EFX in this setting. $\qquad\square$

