,\text{IDO}} = (v_1^{t,\text{IDO}}, \ldots, v_n^{t,\text{IDO}})$.[9] We fix $v_i^{t,\text{IDO}}$ deterministically as follows: sort the multiset $V_i^t$ in non-increasing order (breaking ties arbitrarily but consistently), and assign the $k$-th largest value to good $g_{t+k-1}$ for each $k \in [m - t + 1]$.

Now, consider a single-shot fair division instance with goods $\{g_t, \ldots, g_m\}$. Let $\pi^t = (a_t, \ldots, a_m)$ be a *picking sequence* over goods $\{g_t, \ldots, g_m\}$, whereby $a_i \in N$ for each $i \in \{t, \ldots, m\}$, and agents take turns (according to $\pi^t$) picking their highest-valued good (according to their valuation function) from among the remaining goods available, breaking ties in a consistent manner (say, in favor of lower-indexed goods). Then, given a valuation profile $\mathbf{v} = (v_1, \ldots, v_n)$ and picking sequence $\pi^t$, we denote $\mathcal{A}^{\pi^t, \mathbf{v}} = \left( A_1^{\pi^t, \mathbf{v}}, \ldots, A_n^{\pi^t, \mathbf{v}} \right)$ as the allocation returned in the single-shot instance over the set of goods $\{g_t, \ldots, g_m\}$ induced by the picking sequence $\pi^t$ and valuation profile $\mathbf{v}$.

We then obtain the following result, which will be useful later on.

**Lemma D.1.** *Fix $t \in [m]$. For any picking sequence $\pi^t$ and any valuation profiles $\mathbf{v} = (v_1, \ldots, v_n)$ and $\mathbf{v}^{t,\text{IDO}} = (v_1^{t,\text{IDO}}, \ldots, v_n^{t,\text{IDO}})$ such that for each $i \in N$, $v_i$ and $v_i^{t,\text{IDO}}$ are both consistent with $V_i^t$, we have that*

$$v_i \left( A_i^{\pi^t, \mathbf{v}} \right) \geq v_i^{t,\text{IDO}} \left( A_i^{\pi^t, \mathbf{v}^{t,\text{IDO}}} \right).$$

---

[9]IDO is shorthand for *identical ordering*, as is standard in the literature.

*Proof.* Consider any $t \in [m]$. Suppose we have a picking sequence $\pi^t$ and valuation profiles $\mathbf{v} = (v_1, \ldots, v_n)$ and $\mathbf{v}^{t,\text{IDO}} = (v_1^{t,\text{IDO}}, \ldots, v_n^{t,\text{IDO}})$ such that for each $i \in N$, $v_i$ and $v_i^{t,\text{IDO}}$ are both consistent with $V_i^t$.

An equivalent way to express a picking sequence $\pi^t$ is by looking at the *turns* each agent gets to pick a good. Note that for goods $\{g_t, \ldots, g_m\}$, there will be a total of exactly $m - t + 1$ turns (where at each turn, an agent gets to pick his favorite remaining good).

We introduce several notation:

- Let $\text{turns}_i \subseteq [m - t + 1]$ be the set of turns whereby agents $i$ gets to pick the good under $\pi^t$. Note that $\bigcup_{i \in N} \text{turns}_i = [m - t + 1]$ and for any $i \neq j$, $\text{turns}_i \cap \text{turns}_j = \varnothing$.

- Let $\text{top}_i(k)$ be the $k$-th largest element (counting multiplicity) of the multiset $V_i^t$, for $k \in [m - t + 1]$.

- Let $h_{k,\mathbf{v}}$ be the $k$-th good that was picked according to $\pi^t$ under $\mathbf{v}$.

Consider any agent $i \in N$. Note that for each $k \in \text{turns}_i$ (i.e., in the $k$-th entry of $\pi^t$, it is agent $i$'s turn to pick), under $\mathbf{v}^{t,\text{IDO}}$, since goods in $\{g_t, \ldots, g_m\}$ are identically ordered according to all agents' valuations, agent $i$ will get to (and must) pick her $k$-th favorite good, i.e., $v_i^{t,\text{IDO}}(h_{k,\mathbf{v}^{t,\text{IDO}}}) = \text{top}_i(k)$. Summing over all such $k \in \text{turns}_i$, we get

$$v_i^{t,\text{IDO}}\left(A_i^{\pi^t, \mathbf{v}^{t,\text{IDO}}}\right) = \sum_{k \in \text{turns}_i} v_i^{t,\text{IDO}}(h_{k,\mathbf{v}^{t,\text{IDO}}}) = \sum_{k \in \text{turns}_i} \text{

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

$$|v_1^{1,\mathsf{IDO}}(B_1) - v_1^{1,\mathsf{IDO}}(B_2)| = |1 - 2v_1^{1,\mathsf{IDO}}(B_2)| < |1 - 2a| = |v_1^{1,\mathsf{IDO}}(A_1') - v_1^{1,\mathsf{IDO}}(A_2')|,$$

contradicting the choice of $\{A_1', A_2'\}$ as a minimizer of $|v_1^{1,\mathsf{IDO}}(X) - v_1^{1,\mathsf{IDO}}(Y)|$.

In the remaining (boundary) case, we have $v_1^{1,\mathsf{IDO}}(B_2) = 1 - a$ and $v_1^{1,\mathsf{IDO}}(B_1) = a$, so $\mathcal{B}$ is also a minimizer of the absolute difference. Moreover, $|B_1| = |A_1| + 1 = |A_1'| + 1$ (using (29)), and $B_1$ is the smaller-valued bundle for agent 1 (because $a < 1 - a$). This contradicts the tie-breaking rule that, among difference-minimizing partitions, maximizes the cardinality of the smaller-valued bundle.

Therefore, (32) is impossible and agent 1 satisfies EFX. Thus, $\mathcal{A}$ is EFX. $\qquad\square$

**Theorem 5.4.** *For $n \geq 3$, given frequency predictions, there does not exist any online algorithm with a competitive ratio strictly larger than $0$ with respect to* EFX.

*Proof.* Suppose for a contradiction that there exists a $\gamma$-competitive algorithm for approximating EFX when $n \geq 3$ and with frequency predictions, for some $\gamma \in (0, 1]$. Let $\varepsilon > 0$ be a small constant and $K > 0$ be sufficiently large. Let $V_i = \{K^2, \ldots, K^2, K, \varepsilon, \varepsilon\}$ be the multiset with $n + 1$ elements for each $i \in N$. Let the $n + 1$ goods be $g_1, \ldots, g_{n+1}$ with valuations as follows:

| $\mathbf{v}$ | $g_1$ | $g_2$ | $g_3$ | $g_4$ | $\cdots$ | $g_n$ | $g_{n+1}$ |
|---|---|---|---|---|---|---|---|
| 1 | $K$ | $K^2$ | $\varepsilon$ | $K^2$ | $\cdots$ | $K^2$ | $\varepsilon$ |
| 2 | $K$ | $\varepsilon$ | $\varepsilon$ | $K^2$ | $\cdots$ | $K^2$ | $K^2$ |
| $\vdots$ | $\vdots$ | $\vdots$ | $\vdots$ | $\vdots$ | $\cdots$ | $\vdots$ | $\vdots$ |
| $n$ | $K$ | $\varepsilon$ | $\varepsilon$ | $K^2$ | $\cdots$ | $K^2$ | $K^2$ |

Suppose w.l.o.g. that $g_1$ is allocated to agent $1$. We split our analysis into two cases.

**Case 1: $g_2$ is allocated to agent $1$.** Then no goods in $\{g_3, \ldots, g_{n+1}\}$ can be allocated to agent 1 (otherwise there must exist some agent in $N \setminus \{1\}$ that receives no goods, leading to a 0-EFX allocation). Moreover, if some agent in $N \setminus \{1\}$ receives two goods from $\{g_3, \ldots, g_{n+1}\}$, then there must exist some agent other agent from $N \setminus \{1\}$ that receives no goods, leading to a 0-EFX allocation. Thus, we must have that each good in $\{g_3, \ldots, g_{n+1}\}$ is allocated to a different agent in $N \setminus \{1\}$. Let the agent that is allocated $g_3$ be $i$. Then, $v_i(A_i) = \varepsilon$ and $v_i(A_1 \setminus \{g_2\}) = K$. This gives us $\gamma \leq \frac{\varepsilon}{K} \to 0$ as $K \to \infty$, a contradiction.

**Case 2: $g_2$ is not allocated to agent** 1. Suppose w.l.o.g. that $g_2$ is allocated to agent 2 instead. If at least one of $\{g_3, \ldots, g_{n+1}\}$ is allocated to agent 2 and agent 1 is not allocated any good from $\{g_4, \ldots, g_n\}$, then $v_1(A_1) \leq K + 2\varepsilon$ and $v_1(A_2 \setminus \{g\}) \geq K^2$ for $g \in \arg\min_{g' \in A_2} v_1(g')$. This gives us

$$\gamma \leq \frac{K + 2\varepsilon}{K^2} = \frac{1}{K} + \frac{2\varepsilon}{K^2} \to 0$$

as $K \to \infty$, a contradiction. Therefore, if at least one of $\{g_3, \ldots, g_{n+1}\}$ is allocated to agent 2, agent 1 must be allocated some good from $\{g_4, \ldots, g_n\}$. However, this means that there are at most $n - 3$ remaining goods and $n - 2$ agents, leaving some agent in $N \setminus \{1, 2\}$ with an empty bundle, leading to a 0-EFX allocation. Thus, none of the goods in $\{g_3, \ldots, g_{n+1}\}$ can be allocated to agent 2. This implies that there must be some agent $i \in N \setminus \{2\}$ that receives two goods eventually. However, $v_2(A_2) = \varepsilon$ and $v_2(A_i \setminus \{g\}) \geq K$ for $g \in \arg\min_{g' \in A_i} v_2(g')$. This gives us $\gamma \leq \frac{\varepsilon}{K} \to 0$ as $K \to \infty$, a contradiction.

We arrived at a contradiction in both cases, thereby proving our claim. $\qquad\square$

**Theorem 5.5.** *Let $s$ be a feasible share. There exists an online algorithm that, given the predicted multisets $(\widehat{V}_i)_{i \in N}$, outputs an allocation $\mathcal{A}$ such that for every agent $i \in N$, $v_i(A_i) \geq (1 - \varepsilon_i) \cdot s(\widehat{V}_i, n)$, where $\varepsilon_i \in [0, 1]$ is the relative instantiation error induced by the algorithm's online instantiation rule.*

*Proof.* Fix the predicted multisets $(\widehat{V}_i)_{i \in N}$. The algorithm maintains, for each agent $i$, an online instantiation rule; let $\tilde{v}_i$ be the induced virtual valuation and let $\eta_i$ be the corresponding error.

On each arrival $g_t$, the algorithm first instantiates virtual values $(\tilde{v}_i(g_t))_{i \in N}$ using the maintained multisets $(\widehat{V}_i^t)_{i \in N}$, and then allocates $g_t$ by running the online algorithm whose existence is guaranteed by Theorem 5.1 on the virtual instance $(N, G, (\tilde{v}_i)_{i \in N})$ with frequency predictions $(\widehat{V}_i)_{i \in N}$.

Let $\mathcal{A} = (A_1, \ldots, A_n)$ be the resulting final allocation.

By construction, for each agent $i$, the frequency multiset of $\tilde{v}_i$ is exactly $\widehat{V}_i$. Applying Theorem 5.1 to the virtual instance gives us, for every agent $i \in N$,

$$\tilde{v}_i(A_i) \geq s(\widehat{V}_i, n). \tag{33}$$

Now fix any agent $i \in N$. Using the inequality $x \geq y - |x - y|$ (valid for all real numbers $x, y$), we have

$$v_i(A_i) = \sum_{g \in A_i} v_i(g) \geq \sum_{g \in A_i} \left( \tilde{v}_i(g) - \left| v_i(g) - \tilde{v}_i(g) \right| \right)$$
$$= \tilde{v}_i(A_i) - \sum_{g \in A_i} \left| v_i(g) - \tilde{v}_i(g) \right|.$$

Since $A_i \subseteq G$,

$$\sum_{g \in A_i} \left| v_i(g) - \tilde{v}_i(g) \right| \leq \sum_{t=1}^{m} \left| v_i(g_t) - \tilde{v}_i(g_t) \right| = \eta_i.$$

Combining this with (33) gives us

$$v_i(A_i) \geq s(\widehat{V}_i, n) - \eta_i. \tag{34}$$

Now, if $s(\widehat{V}_i, n) = 0$, then $v_i(A_i) \geq 0$ holds because valuations are nonnegative. Thus, we