# OpenReview forum: "Online Fair Division with Additional Information"
_ICML.cc/2026/Conference — ICML 2026 regular_

### Official Review · Reviewer_jd2E · 2026-03-03

**Soundness:** 3
**Presentation:** 4
**Significance:** 2
**Originality:** 3
**Overall Recommendation:** 4
**Confidence:** 4

**Summary:**

The paper studies the problem of allocating items to agents in an online setting. The authors focus on fairness notions such as EFX (envy freeness after removing any item), EF1 (envy freeness after removing one item), PROP1 (receiving the proportional value up to one item), and MMS (the maximin share). The main difference from prior work is that the authors consider different degrees of knowledge of future items.

The first results however are in the absence of future information, specifically the authors prove a strong impossibility result on EF1 an an impossibility result on PROP1 under certain conditions.

The positive results begin with the case when the total sum of valuations is known. Algorithm 1 tracks an "outside-max" quantity to achieve PROP1 and EF1 for 2 agents. Another impossibility result however is shown for EFX in this setting.

When the possible incoming values are known, the authors design an algorithm that translates offline fairness guarantees to the online setting. This achieves a new MMS bound for any number of agents, and EFX for 2 agents. Finally the results are extended into a learning augmented framework.

**Compliance With Llm Reviewing Policy:**

Affirmed.

**Final Justification:**

The authors have addressed the points I raised in my initial review. Specifically, their explanation that the 2-agent limitation for EF1 is a structural boundary, rather than a technical shortcoming, is convincing, as their impossibility results complete the theoretical picture. The authors also provided a solid justification for the necessity of additive valuations in the positive results. Weighing the dimensions, the submission is technically sound, the presentation is clear, and it provides a good spin on a traditional setting. Finally, the rebuttal effectively addressed the concerns from other reviewers regarding the prediction models and algorithmic comparisons. My assessment remains positive, and I maintain my recommendation to accept the paper.

**Key Questions For Authors:**

Are extensions to other valuation functions possible?

What are the barriers that do not allow us to go beyond 2 agents in your techniques?

**Limitations:**

Yes

**Strengths And Weaknesses:**

Soundness

Strengths:
- The proofs are correct and the authors even fix an earlier mistake in the literature.

Presentation

Strengths:
- The presentation is clear. It was easy to go through the paper

Significance

Strengths:
- The paper studies a fundamental problem and finds interesting new questions in it
- Nicely motivated by ML applications

Weaknesses:
- Most of the positive results focus on the limited 2 agent case
- No consideration of valuation functions beyond additive

Originality

Strengths:
- It is a good spin on a traditional setting.

---

> ### Author Rebuttal · Authors · 2026-03-29
>
> Thank you for your thoughtful and positive review. Please find our response below.
>
> ---
>
> > Most of the positive results focus on the limited 2 agent case. What are the barriers that do not allow us to go beyond 2 agents in your techniques?
>
>
> For envy-based notions, the barrier is structural rather than a limitation of our techniques.
> Under normalization information, Theorem 4.4 shows that for $n \ge 3$, no online algorithm can guarantee any positive approximation to EF1, while Theorem 4.3 already rules out any positive approximation to EFX even for $n=2$.
> Under frequency predictions, Theorem 5.4 again shows a zero competitive ratio for EFX when $n \ge 3$. So the $n=2$ vs. $n \ge 3$ transition is itself one of the paper’s main findings.
> At the same time, the paper does obtain general $n$ positive guarantees: PROP1 for all $n$ under normalization, and offline best share-based guarantees (including the current best MMS approximation) for all $n$ under frequency predictions.
>
> ---
>
> > No consideration of valuation functions beyond additive. Are extensions to other valuation functions possible?
>
> Our negative results are already robust, because they are proved on additive instances and therefore immediately extend to any broader valuation class containing additive valuations.
>
> The positive side, however, uses additivity in an essential way. For Algorithm 1, $v_i(A_i)+\frac{n-1}{n}M_i$ relies on reasoning about bundle values plus a single outside good. For frequency predictions, the model itself summarizes future information via per-agent scalar multisets and the IDO construction. These are specifically additive mechanisms, so extending the positive results to submodular or subadditive valuations would require genuinely new ideas.
>
> This is also consistent with the offline fair division literature: EF1 itself continues to make sense and is known to exist even under general monotone/combinatorial valuations (Lipton et al., 2004), but more recent work shows that additive-style envy guarantees do not extend cleanly to submodular settings and motivates new notions/techniques there (Caragiannis et al., 2019; Montanari et al., 2025). Share-based fairness beyond additivity is also studied with different tools, e.g., via MMS/APS-style guarantees for submodular valuations (Ben Uziahu and Feige, 2023).

---

> > ### Author Rebuttal · Reviewer_jd2E · 2026-04-02
> >
> > Thank you for the responses, they address the points that I raised. My score remains positive.

---

### Official Review · Reviewer_Tsfj · 2026-03-10

**Soundness:** 4
**Presentation:** 3
**Significance:** 3
**Originality:** 3
**Overall Recommendation:** 5
**Confidence:** 3

**Summary:**

The paper studies the problem of online fair division with adversarial item arrivals and additive valuations under different fairness notions, where the goal is to obtain ex-post competitive ratios (CR). The main results are as follows:

1. Under no information about the future, no CR for EF1 and PROP1 is possible, even with only two agents.
2. Under the normalization assumption (the values of all agents sum up to 1), the authors provide a novel algorithm that achieves PROP1 for any number of agents and EF1 for two agents. They also show tight impossibility results: for more than two agents, EF1 is impossible to achieve, and even with normalization, EFX cannot be achieved.
3. If frequency predictions are available as an input assumption, then any share-based fairness notion that is achievable in the single-shot setting can also be obtained via an online algorithm with frequency predictions. This directly allows the extension of \(3/4\)-MMS approximations to this setting, which is natural given that ordered instances are the hardest cases. More importantly, they show that EFX is achievable with two agents and impossible to achieve with more than two agents.

Finally, the authors connect their results to learning-augmented predictions and show that the guarantees are robust and degrade gracefully with the error parameters. They also show that with identical valuations, EF1 is achievable without any further information.

**Compliance With Llm Reviewing Policy:**

Affirmed.

**Key Questions For Authors:**

1- What is an IDO instance? It is mentioned without a proper definition.

2- In the frequency-prediction framework, can the authors give a short worked example of the IDO-instance construction and the induced online picking sequence?

**Limitations:**

yes

**Strengths And Weaknesses:**

Strengths:

- The paper studies an important problem of online fair division under item arrivals with additional assumptions, across different fairness notions such as PROP1, EFX, EF1, and maximin share.
- It provides an almost complete picture of possibilities and impossibilities under two natural assumptions: normalization and frequency predictions.
- I also like the Algorithm 1 story. The “outside-good certificate” is a simple but nontrivial device, and the paper explains the intuition well.

Weaknesses:

- Algorithm 2 is important and should be included in the main body, together with some intuitive explanation. Overall, the presentation of the frequency-prediction framework would benefit from a clearer explanation of some of its central concepts and constructions.
- I believe it would be nice to add the last missing piece: EF1 for more than two agents under frequency predictions. Either an algorithm or an impossibility result for this case would complete the picture.

---

> ### Author Rebuttal · Authors · 2026-03-29
>
> Thank you for your thoughtful and very positive review! Please find our response below.
>
> ---
>
> > I believe it would be nice to add the last missing piece: EF1 for more than two agents under frequency predictions. Either an algorithm or an impossibility result for this case would complete the picture.
>
> We investigated it extensively, but do not yet have a proof either way: it is a challenging problem.
> We will strengthen the paper's discussion of this point by explaining more explicitly why Theorem 5.1 does not automatically resolve EF1: the theorem applies to share-based guarantees, whereas EF1 is a pairwise envy-based notion and is not directly captured by that framework.
>
> ---
>
> > What is an IDO instance? It is mentioned without a proper definition.
>
> IDO stands for identical ordering: given the remaining multiset $V_i^t$ for each agent $i$, we sort it in non-increasing order and assign the k-th largest value to the $k$-th remaining good $g_{t+k-1}$.
> Thus, all agents rank the remaining goods in the same order. We will define this at first use.
>
> ---
>
> > In the frequency prediction framework, can the authors give a short worked example of the IDO-instance construction and the induced online picking sequence?
>
> For instance, if $V_1=set(7,4,1)$ and $V_2=set(6,5,1)$, then the induced IDO instance on goods $g_1,g_2,g_3$ is
> $v_1^{IDO}=(7,4,1)$ and $v_2^{IDO}=(6,5,1)$,
> so both agents rank $g_1 \succ g_2 \succ g_3$.
> If an offline share-achieving allocation on this IDO instance is $X_1=\{g_1,g_3\}$ and $X_2=\{g_2\}$, then the induced picking sequence is $(1,2,1)$: the first pick gives $g_1$ to agent $1$, the second gives $g_2$ to agent $2$, and the third gives $g_3$ to agent $1$. We will add exactly this kind of example to make understanding Algorithm 2 more intuitive.

---

> > ### Author Rebuttal · Reviewer_Tsfj · 2026-04-03
> >
> > Thank you for the clarifications. In the fair division literature, I've always seen 'ordered instances' referred to the instances the authors call IDO, so I suggest using this term. I will keep my positive score.

---

### Official Review · Reviewer_sGRq · 2026-03-12

**Soundness:** 3
**Presentation:** 3
**Significance:** 3
**Originality:** 3
**Overall Recommendation:** 4
**Confidence:** 3

**Summary:**

This paper studies the problem of online fair division of indivisible goods, where a fixed set of agents must allocate items that arrive sequentially and must be assigned irrevocably upon arrival. The setting captures uncertainty about future items, as the algorithm observes the arrival of goods one by one along with the agents’ valuations and must make allocation decisions without knowledge of future arrivals. The work focuses on classical fairness notions such as proportionality, envy-freeness, and maximin share fairness, along with their common relaxations, including EF1, EFX, PROP1, and approximate MMS guarantees. The goal is that the final allocation obtained after the sequential arrival of goods satisfies these fairness criteria. The authors further provide both algorithmic results and impossibility results for several fairness notions in the online setting. In particular, under normalised valuation assumptions, the authors propose an algorithm that achieves EF1 for n=2 and PROP1 for general n. In addition, the paper studies an information model based on frequency predictions, where a meta-algorithm is proposed that lifts a broad class of offline share-based fairness guarantees to the online setting, matching the best known offline bounds. The paper further considers a learning-augmented setting, providing robustness guarantees when predictions are noisy, including cases with noisy totals or imperfect frequency predictions.

**Compliance With Llm Reviewing Policy:**

Affirmed.

**Key Questions For Authors:**

1) Can authors clarify what the key algorithmic differences between the proposed algorithm and the approach in Zhou et al. (2023) are?

2) Many of the positive guarantees rely on the assumption that valuations are normalized and the total value of items is known in advance.

a) How sensitive are the results to this assumption?

b) Do the guarantees degrade if the total value is only approximately known or estimated?

3) While the paper is theoretical, simple simulations could help illustrate the algorithm’s behavior. Have the authors considered running small simulations to illustrate how the algorithms behave under realistic arrival sequences or prediction errors?

**Limitations:**

1) Adding experimental analysis will be helpful as this paper is in learning setting which is a very practical application.

2) Providing the tightness guarantees for the proposed algorithms will be useful

3) Justification on normalised valuation will be helpful.

**Strengths And Weaknesses:**

1) Soundness: The paper is theoretical in nature and provides complete proof of the claims presented in the paper. To the best of my understanding, the proofs look correct to me. Since this is a theoretical paper, there is no experimental analysis. The work provides theoretical guarantees in the online setting, including EF1 for two agents and PROP1 for arbitrary n under normalized valuations. These results clarify which fairness notions remain achievable despite the constraints of online allocation. However, the robustness guarantees under noisy predictions remain somewhat unclear. While the paper discusses robustness in the learning-augmented setting and considers noise in totals and frequency predictions, it does not provide explicit bounds quantifying how prediction errors affect the final fairness guarantees. In particular, it would be useful to understand how the approximation ratios degrade as a function of the prediction error. Providing formal robustness bounds or tighter error-dependent guarantees would strengthen the theoretical understanding of the proposed approach. Although experiments are not always necessary for theoretical work, simulation results illustrating the behaviour of the algorithms under realistic arrival patterns or prediction errors could provide additional intuition and strengthen the empirical relevance of the paper.

2) Presentation: The paper reads well and is easy to follow. The paper discusses some work in the bandit feedback setup, and it is not clear how the paper itself relates to this setting. However, the algorithmic novelty of Algorithm 1 and the role of normalization are not sufficiently explained. The algorithm is presented as a modification and generalization of the semi-online algorithm of Zhou et al. (2023), which already considers online allocation with sequential arrivals and additive normalized valuations. However, the paper does not clearly describe the precise algorithmic differences or why the modifications are necessary. Both approaches share several structural similarities, including greedy allocation, maintaining active agents, and deactivating agents once a fairness threshold is reached. While the paper claims stronger guarantees (EF1 for n =2 and PROP1 for general n), the mechanism enabling these guarantees is not clearly explained. In particular, the role of the fairness score xi and the threshold 1/n is not intuitively motivated, nor is it clear how normalization interacts with this score to approximate proportional fairness. A clearer comparison with Zhou et al. (2023) would strengthen the presentation.

3) Significance: The paper studies online fair division of indivisible goods with sequential arrivals and irrevocable allocations. Ensuring fairness under uncertainty about future items is a fundamental challenge in algorithmic fairness and online resource allocation. However, Several positive fairness guarantees rely heavily on the normalized valuation assumption, where the total value of all items is known in advance. While this assumption is common in semi-online models, which may be unrealistic in many practical settings where the total value of future items is unknown. The frequency prediction model relies on strong assumptions and limited algorithmic novelty. In particular, it assumes access to exact frequency predictions of future valuation values, effectively revealing the distribution of valuations and even the total number of future items, which may be unrealistic in many practical online settings. Moreover, the fairness guarantees obtained in this setting are largely inherited from existing offline results. Algorithm 2 essentially simulates an offline picking sequence using the predicted valuation frequencies, and thus primarily acts as a meta-framework that transfers offline share guarantees to the online setting rather than introducing a fundamentally new allocation strategy or improving known fairness approximation bounds.

4) Originality: The frequency prediction model shows how offline share-based fairness guarantees can be lifted to the online setting using predictive information. This provides a useful conceptual bridge between offline and online fair division frameworks. The paper analyzes robustness under noisy predictions. The framework highlights how predictive information can improve fairness guarantees while maintaining tolerance to bounded prediction errors.

---

> ### Author Rebuttal · Authors · 2026-03-29
>
> Thank you for your thoughtful and positive review. Please find our response to each point below.
>
> ---
>
> **Bandit feedback.**
> We discussed this distinction in Sec 1.2: Recent bandit-based online fair division papers study uncertainty about values or distributions, limited/noisy feedback, and regret-style or fairness in expectation objectives over time (e.g., Yamada et al., 2024; Procaccia et al., 2024).
> In contrast, our model observes the full valuation vector of the current item, allows adversarial arrival order, and studies ex post fairness of the final allocation. So while the themes are adjacent, the information structure/guarantees are fundamentally different.
> We will clarify this further.
>
> ---
>
> **Algorithmic difference from Zhou et al. (2023).**
> The (algorithmic) comparison with Zhou et al. (2023) was addressed in Section 1.1, but we are happy to restate the key algorithmic difference more explicitly.
> Zhou et al.’s two-agent goods result explicitly targets a $0.5$-competitive MMS guarantee under normalization.
> Our Algorithm 1 is instead built around the certificate $x_i = v_i(A_i) + \frac{n-1}{n} M_i$, where $M_i$ is the value of the largest outside good for agent $i$.
> The point of this certificate is exactly that once $x_i \geq 1/n$, agent $i$ already has a PROP1 witness, and in the $n=2$ case this gives an EF1 witness.
> So the novelty is not merely “greedy allocation with deactivation”, but the specific outside good certificate and threshold analysis tailored to PROP1/EF1.
> As noted as well, Zhou et al.’s procedure can fail PROP1, and thus EF1, even when $n=2$.
>
> ---
>
> **Robustness/error dependence.**
> In Section 4.1 and Theorem 4.5, we provide explicit bounds quantifying how prediction errors affect the final fairness guarantees in the submission.
> We proved that with certified intervals $[\underline T_i,\overline T_i]$, the guarantee degrades gracefully: for $n=2$, EF1 degrades additively by $\rho_i=\overline T_i-\underline T_i$, and for general $n$, PROP1 degrades multiplicatively by $\kappa_i=\underline T_i/\overline T_i$.
> The theorem also states that EF1 degrades linearly in $\rho_i$, while PROP1 degrades multiplicatively via $\kappa_i$.
> In Appendix C.1, we explain why an additive robustness notion is the natural one for EF1 in this model.
>
> The same holds for noisy frequency predictions. Section 5.1 and Theorem 5.5 already give the bound $v_i(A_i)\ge (1-\varepsilon_i) s(\widehat V_i,n)$, where $\varepsilon_i$ is the relative instantiation error. The paragraph immediately after Theorem 5.5 states the standard learning-augmented guarantees of consistency, robustness, and smoothness, and further explains the uniform-error specialization $v_i(A_i)\ge (1-\varepsilon)s(\widehat V_i,n)$. When $s$ is an $\alpha$-approximation to a benchmark such as MMS, the result is already stated as an additional multiplicative $(1-\varepsilon_i)$ factor on top of $\alpha$.
>
> ---
>
> **Realism of normalization/frequency predictions.**
> We agree that exact advice is an abstraction, but it is a standard and useful one in semi-online and learning-augmented algorithm design.
> More importantly, the paper does not treat advice as "all or nothing"; Theorems 4.5 and 5.5 already analyze noisy normalization information and noisy frequency predictions.
> On the modeling side, frequency predictions has clear precedents; Im et al. (2021) study online knapsack with very weak predictions given by upper/lower bounds on the number of items of each value; Angelopoulos et al. (2023) study online bin packing with possibly erroneous frequency predictions and explicitly motivate them as realistic learnable predictions; Balseiro et al. (2023) model revenue management advice as a frequency table of future fare classes; and Boyar et al. (2017) survey the broader advice/semi-online viewpoint.
>
> ---
>
> **Novelty of Sec 5.**
> We respectfully disagree that this diminishes novelty.
> A constructive transfer theorem might be as (if not more) valuable than a bespoke algorithm for one fairness notion, because it identifies the correct abstraction and gives us multiple corollaries at once. Here, the nontrivial content is exactly that an offline share-achieving allocation on the IDO surrogate can be converted into a feasible online sequence of irrevocable choices under only marginal frequency advice. Algorithm 2 is not simply replaying an offline sequence with access to the future; it proves that such a sequence can be realized online for every stream consistent with the predictions. Matching offline share guarantees online under strictly weaker than full information is one of our contribution.
>
> ---
>
> **Experiments.**
> We appreciate the suggestion. Since the paper’s contribution is a worst-case structural characterization with exact positive and impossibility results, we prioritized complete proofs over experiments. We do agree, however, that a short worked example can improve intuition for Algorithm 2 and the noisy prediction model, and we will add one.

---

> > ### Author Rebuttal · Reviewer_sGRq · 2026-04-04
> >
> > I am convinced by the response provided by the authors.

---

### Official Review · Reviewer_Damb · 2026-03-12

**Soundness:** 3
**Presentation:** 4
**Significance:** 3
**Originality:** 2
**Overall Recommendation:** 4
**Confidence:** 4

**Summary:**

The paper studies an interesting problem, with well-motivated fairness objectives of approximate EF1, EFX, and PROP1 for the online allocation of indivisible goods under various information models.

**Compliance With Llm Reviewing Policy:**

Affirmed.

**Final Justification:**

I would like to thank the authors for their comprehensive response. The clarifications provided have addressed my remaining concerns, and I am pleased to maintain my positive evaluation of the work.

**Key Questions For Authors:**

n/a

**Limitations:**

The author should clarify the difference between your model with other existing related concepts, e.g., prophet inequality.

**Strengths And Weaknesses:**

Strengths

The authors provide matching impossibility results for almost all of their positive algorithmic guarantees. For example, they prove that with normalization information, EF1 is achievable for $n=2$, but definitively prove it is impossible for $n\ge3$. This cleanly establishes the exact limits of what specific advice models can achieve. For frequency predictions, instead of designing one-off algorithms for specific fairness notions, they provide a constructive meta-algorithm that lifts any feasible offline "share-based" guarantee to the online setting (Theorem 5.1).

Weakness

1. For normalization information, the EF1 guarantee applies to only two agents. The authors admit that no positive approximation to EF1 is possible for three or more agents. A fairness guarantee that exclusively works for $n=2$ is not particularly appealing for generalized multi-agent systems.
2. The conceptual contribution of "frequency predictions" is overly strong. By assuming the algorithm knows the exact multiset of future values (just lacking the arrival order), the model effectively strips away the core uncertainty of online division. It is entirely unsurprising that providing the exact multiset of values allows one to recover offline "share-based" guarantees.

---

> ### Author Rebuttal · Authors · 2026-03-29
>
> Thank you for your thoughtful and positive review. Please find our response below.
>
> ---
>
> > For normalization information, the EF1 guarantee applies to only two agents. The authors admit that no positive approximation to EF1 is possible for three or more agents. A fairness guarantee that exclusively works for $n=2$ is not particularly appealing for generalized multi-agent systems.
>
> We agree that an $n=2$ result would be limited if it stood in isolation.
> Our work, however, shows precisely that it does not stand in isolation.
> Under normalization information, Theorem 4.2 gives PROP1 for all $n$ (which is also a natural relaxation of EF1 studied in the offline fair division literature), Theorem 4.1 gives exact EF1 for $n=2$, and Theorem 4.4 shows a matching impossibility showing that for $n \ge 3$, no online algorithm can guarantee any positive approximation to EF1.
> Thus, the restriction to two agents for EF1 is not a limitation of our proof technique but a structural boundary of the model.
> In that sense, our contribution is basically a characterization of what normalization information can and cannot buy in this setting.
> Moreover, the paper also contain other general $n$ positive results, e.g., offline best share-based guarantees for all $n$ under frequency predictions.
>
> ---
>
> > The conceptual contribution of ``frequency predictions'' is overly strong. By assuming the algorithm knows the exact multiset of future values (just lacking the arrival order), the model effectively strips away the core uncertainty of online division. It is entirely unsurprising that providing the exact multiset of values allows one to recover offline "share-based" guarantees.
>
> We thank the reviewer for raising this concern.
> We addressed this point in Section 5: the predictor does not reveal the future item-value vectors; it reveals only for each agent separately, the multiset of scalar values that will appear.
> In particular, it does not reveal the arrival order, it does not reveal which future item realizes which value, and it does not reveal the cross-agent coupling of values on the same item.
> In particular, that missing joint structure is exactly what keeps the problem online.
> A concrete way to see this is the following: with two agents and two future goods, the same predicted multisets: $V_1 = V_2 = \{1,0\}$ are both consistent with both valuation vectors $((1,1),(0,0))$ and $((1,0),(0,1))$.
> These two futures have identical per-agent marginals but very different allocation structure. Thus the advice is strictly weaker than seeing the future sequence.
> This is also why Theorem 5.1 is not a entirely unsurprising corollary of offline existence: its contribution is a constructive online lifting theorem that turns an offline share-achieving allocation on the induced IDO instance into an irrevocable online policy that works for every realization consistent with only these marginal forecasts.
> Frequency predictions are also well-studied in online algorithms, including online knapsack, bin packing, and revenue management with advice (Boyar et al., 2017; Im et al., 2021; Angelopoulos et al., 2023; Balseiro et al., 2023).
>
>
> ---
>
> > The author should clarify the difference between your model with other existing related concepts, e.g., prophet inequality.
>
> Our model is closer to semi-online/advice and learning-augmented models than to prophet inequalities.
> Prophet inequalities classically arise from stochastic optimal stopping/selection problems benchmarked against a prophet, whereas our setting is adversarial, multi-agent, and evaluates fairness _ex post_ after every arriving item has been allocated irrevocably.
> That is why the more natural comparison class is advice/semi-online and learning-augmented online algorithms, not prophet inequalities.
> But we will clarify this, thanks for the suggestion!

---

> > ### Author Rebuttal · Reviewer_Damb · 2026-04-06
> >
> > I would like to thank the authors for their comprehensive response. The clarifications provided have addressed my remaining concerns, and I am pleased to maintain my positive evaluation of the work.

---

### Decision · Program_Chairs · 2026-04-30

**Decision:**

Accept (regular)

**Comment:**

The paper studies online fair division under varying levels of future information, providing a near-complete characterization of which fairness notions are achievable under no information, normalization, and frequency predictions, with matching impossibilities and learning-augmented variants. All four reviewers are positive, finding the paper well-written, technically sound, and providing a near-complete structural picture. I recommend acceptance.